# Half-wormholes in nearly AdS$_2$ holography

**Antonio M. García-García$^{1\star}$ and Victor Godet$^{2\dagger}$**

**1** Shanghai Center for Complex Physics, School of Physics and Astronomy,
Shanghai Jiao Tong University, Shanghai 200240, China
**2** International Centre for Theoretical Sciences (ICTS-TIFR),
Tata Institute of Fundamental Research, Shivakote, Hesaraghatta,
Bangalore 560089, India

$\star$ amgg@sjtu.edu.cn, $\dagger$ victor.godet@icts.res.in

## Abstract

We find half-wormhole solutions in Jackiw-Teitelboim gravity by allowing the geometry to end on a *spacetime D-brane* with specific boundary conditions. This theory also contains a Euclidean wormhole which leads to a factorization problem. We propose that half-wormholes provide a gravitational picture for how factorization is restored and show that the Euclidean wormhole emerges from averaging over the boundary conditions. The wormhole is known to be dual to a Sachdev-Ye-Kitaev (SYK) model with random complex couplings. We find that the free energy of the half-wormhole is strikingly similar to that of a single realization of this SYK model. These results suggest that the gravitational path integral computes an average over spacetime D-brane boundary conditions.

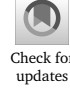

# 1  Introduction

The Euclidean path integral has led to many insights in quantum gravity [1–3] but its true meaning remains elusive. A central question is whether one should include wormholes in the sum over geometries. Spacetime wormholes have been extensively studied in the 80s and have raised numerous conceptual puzzles [4–9]. In the context of the AdS/CFT correspondence [10], they lead to a factorization problem [11].

Recent developments have shown that spacetime wormholes seem to be capturing averaged properties of an exact theory. In Jackiw-Teitelboim gravity [12, 13], the ramp in the spectral form factor is captured by a "double cone" wormhole [14]. Wormholes were crucial in showing that certain correlation of JT gravity are dual to those of an ensemble of random matrices [15–17]. Replica wormholes were used to obtain a unitary Page curve in toy models of evaporating black holes [3, 18, 19] although their relevance in realistic black hole evaporation can be debated [20, 21], and they also appear in the computation of the quenched free energy [22]. Spacetime wormholes have also been studied in three and higher dimensions [23–31].

These developments suggest that the Euclidean path integral is computing some kind of ensemble average. The question is then: what are we averaging over in the gravity theory? In this paper, we propose an answer based on a precise analysis in the context of nearly AdS$_2$ holography [32–34].

Our work extends a possible resolution of the factorization problem proposed in [35] where a single realization of the Sachdev-Ye-Kitaev (SYK) model [32, 36–45] at one point in time was analyzed. It was observed that factorization is restored due to novel saddle-points which

behave as "half-wormholes".[1] However, the analysis of [35], although completely precise, was performed only at one point in time and didn't provide a gravitational picture.

In this paper, we study a more realistic model consisting of JT gravity with a massless scalar field and its dual SYK model. In this theory, the gravitational path integral must be defined as the sum over saddle-points which is closer to what we expect in higher dimensions. Our starting point is the Euclidean wormhole supported by imaginary sources described in [48]. By allowing the geometry to end on some *spacetime D-brane*, or SD-brane[2], we find half-wormhole solutions, characterized by a choice of boundary condition $j(\tau)$ for the scalar field at the small end of the trumpet geometry.

We propose that half-wormholes appear in the "exact", or non-averaged, gravity theory, where the wormhole is excluded so that factorization is manifest. After averaging over the boundary conditions, we recover the Euclidean wormhole solution of the "simple" gravity theory in which factorization is lost. This leads to the general proposal that the Euclidean path integral is an average over SD-brane boundary conditions.

Our gravity setup is especially nice because it has a holographic dual. The Euclidean wormhole was shown in [48] to be dual to a two-site SYK model with complex couplings where it was argued that the wormhole emerges in the disorder average. We thus expect that the "exact" theory with half-wormholes is dual to a single realization of the SYK model.

This expectation is confirmed by the remarkable match of the complex free energies. In particular, we find evidence that the zero mode of $j(\tau)$ should be identified the mean value of the complex SYK coupling as defined in (93).

## 2 Review of the duality

In this section, we give a brief review of the duality described in [48] between a Euclidean wormhole in JT gravity and a two-site SYK model with complex couplings. For lack of a better term, we use here the word "duality" in a loose sense as the two setups are only approximately equivalent; we would like to view the JT gravity story as a toy version of what happens in the true dual of SYK.

### 2.1 Euclidean wormhole

The theory we consider is Jackiw-Teitelboim gravity with matter consisting of a massless scalar field $\chi$. The action is

$$S = S_{\text{JT}} + S_\chi, \tag{1}$$

where

$$
\begin{aligned}
S_{\text{JT}} &= -\frac{S_0}{2\pi}\left[\frac{1}{2}\int d^2x\sqrt{g}R + \int d\tau\sqrt{h}K\right] - \frac{1}{2}\int d^2x\sqrt{g}\,\Phi(R+2) - \int d\tau\sqrt{h}\,\Phi(K-1), \\
S_\chi &= \frac{1}{2}\int d^2x\sqrt{g}\,(\partial\chi)^2.
\end{aligned}
\tag{2}
$$

The Euclidean wormhole solution is given the double trumpet geometry

$$ds^2 = \frac{d\tau^2 + d\rho^2}{\cos^2\rho}, \qquad -\frac{\pi}{2} \le \rho \le \frac{\pi}{2}, \quad \tau \sim \tau + b. \tag{3}$$

---

[1]See also [46, 47] for recent discussions on half-wormholes.

[2]SD-branes have also been discussed in a related context in [49–52].

This solution needs to be supported by imaginary boundary sources for the scalar field. This corresponds to imposing Dirichlet boundary conditions

$$\lim_{\rho \to \pi/2} \chi = ik, \qquad \lim_{\rho \to -\pi/2} \chi = -ik, \tag{4}$$

where $k$ is a positive real number.

The partition function of the wormhole is given by

$$Z_{\text{WH}} = z_{\text{WH}} \, e^{-S_{\text{WH}}}, \tag{5}$$

where the classical action and one-loop prefactor are

$$S_{\text{WH}} = b^2 T - \frac{2bk^2}{\pi}, \qquad z_{\text{WH}} = \frac{T}{2\pi} \frac{1}{\prod_{n \geq 1}(1 - e^{-nb})}. \tag{6}$$

Our regime of interest corresponds to a low temperature regime in which $b \gg 1$ so that the one-loop contribution can be neglected.[3] The on-shell value of $b$ can be determined by extremizing $S_{\text{WH}}$ which leads to

$$b_* = \frac{k^2}{\pi T}, \tag{7}$$

and a constant free energy

$$F_{\text{WH}} = -\frac{k^4}{\pi^2}. \tag{8}$$

We should compare this with the free energy of two black holes

$$F_{\text{BH}} = -2S_0 T - 4\pi^2 T^2, \tag{9}$$

and we see that there is a transition at the critical temperature

$$T_c = \frac{k^4}{2\pi^2 S_0} \qquad (k^2 \ll S_0). \tag{10}$$

This transition is depicted in Fig. 1a.

This computation gives the annealed free energy in gravity. Using a simple replica computation, it was shown in [48] that the annealed and quenched free energies are the same up to negligible corrections.

We have included Appendix A which gives a more complete description of the wormhole in terms of its Schwarzian effective action. Interestingly, the wormhole is entirely a consequence of a U(1) gauge constraint. It is equivalent to two decoupled Liouville particles with correlated potentials, in a way rather similar to [53]. In particular, the energy gap is given as

$$E_{\text{gap}} = \frac{2k^2}{\phi_r \pi^2}. \tag{11}$$

We conclude this summary by commenting about the physical significance of imaginary sources for a real scalar field. In Lorentzian signature, imaginary sources would be unphysical as they lead to violations of the averaged null energy condition. Among other things, they could be used to construct AdS traversable wormholes without a non-local coupling between the boundaries, a violation of the "no-transmission principle" [54] (see [55] for a related discussion). In Euclidean signature, these imaginary sources do make sense. They can be defined by analytic continuation from real sources in the thermal partition function and can be used to study the statistical mechanics of a physical system. For example, imaginary chemical potentials have been useful in studying the phase diagram of gauge theories [56,57]. This is analogous to the spectral form factor which is obtained by analytically continuing the inverse temperature.

---

[3]Note that this contribution becomes important at high temperature and actually leads to another "small wormhole" saddle-point discussed in [48].

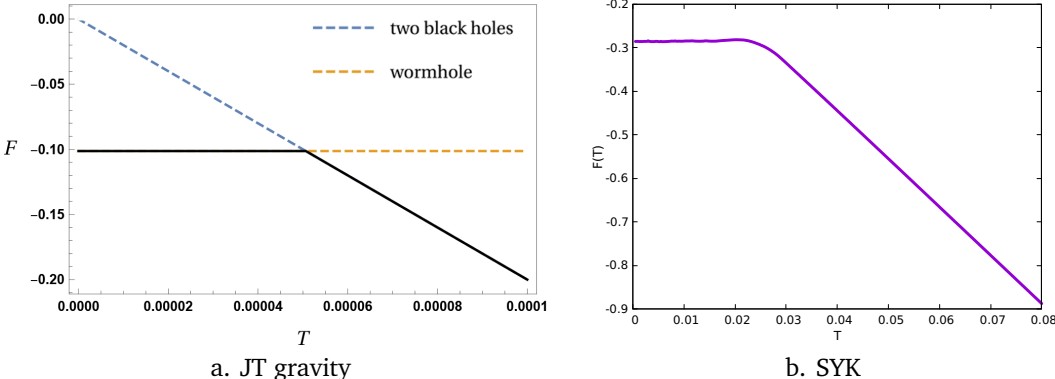

a. JT gravity             b. SYK

Figure 1: Free energy of the JT gravity and the two site complex SYK setup of Ref. [48] for $N = 34$ Majoranas on each site after performing a quench average over 45 disorder realizations. In both cases, we observe a first order phase transition at low temperature.

## 2.2 SYK with complex couplings

The dual setup is a two-site SYK model with Hamiltonian

$$H = H_L + H_R, \tag{12}$$

each of which is an SYK Hamiltonian with complex couplings

$$H_L = \frac{1}{4!} \sum_{i,j,k,\ell=1}^{N/2} (J_{ijk\ell} + i\kappa M_{ijkj\ell}) \psi_i^L \psi_j^L \psi_k^L \psi_\ell^L, \tag{13}$$

$$H_R = \frac{1}{4!} \sum_{i,j,k,\ell=1}^{N/2} (J_{ijk\ell} - i\kappa M_{ijkj\ell}) \psi_i^R \psi_j^R \psi_k^R \psi_\ell^R, \tag{14}$$

where $\kappa$ is a positive real number. The couplings are Gaussian distributed with zero average and standard deviation $\sqrt{\langle J_{ijk\ell}^2 \rangle} = \sqrt{\langle M_{ijk\ell}^2 \rangle} = (12/N)^{3/2}$.

We are interested here in the quenched free energy

$$\langle F \rangle = -T \langle \log Z \rangle, \tag{15}$$

which is plotted as a function of the temperature in Fig. 1b.

## 2.3 JT/SYK duality

We proposed in [48] that the two models described above are dual to each other.[4] By the AdS/CFT dictionary, the imaginary parts of the SYK couplings act as imaginary sources for bulk operators. This suggests that the parameter $\kappa$ should be dual to $k$. Note that the massless scalar field can only be viewed as a toy version of the complicated SYK operator which multiplies $\kappa$. We are not trying to establish an exact duality here: the JT gravity story should be viewed as a simplified version of what happens in the true dual of SYK. Still, we will see that the two systems are remarkably similar.

---

[4]Note that the relation between complex couplings in the SYK model and Euclidean wormholes was anticipated in [53].

The most important check is that the free energies of the two models have exactly the same phase transition at low temperature, see Fig. 1. We can also perform a number of more quantitative checks. In the SYK setup, it was shown in [58] that the critical temperature and energy gap have the following dependence with $\kappa$

$$T_c \sim \kappa^4, \qquad E_{\text{gap}} \sim \kappa^2, \tag{16}$$

from an analysis of the Schwinger-Dyson equations. This perfectly matches the dependence with $k$ of the corresponding quantities in JT gravity given in (10) and (11).

Thus, the massless scalar field in JT gravity is sufficient to reproduce the main features of the imaginary part of the SYK coupling at low temperatures. Note that this imaginary part corresponds to a deformation of the SYK Hamiltonian by the operator

$$\frac{1}{4!} \sum_{i,j,k,\ell=1}^{N/2} M_{ijkj\ell} \psi_i^L \psi_j^L \psi_k^L \psi_\ell^L, \tag{17}$$

which is a complicated marginal operator with $\Delta = 1$. Although the gravity dual of SYK is not well understood, the holographic dictionnary says on general grounds that this deformation should be dual to adding boundary sources for a massless scalar field $\chi$ and we have seen that this indeed gives an adequate model.

Remarkably, we will see in this work that this JT/SYK duality seems to extend to a single realization of the SYK couplings.

## 3 Half-wormhole solution

The theory we consider is Jackiw-Teitelboim gravity with a massless scalar field $\chi$ as described by the action (1).

### 3.1 Geometry and boundary conditions

The half-wormhole is described by the trumpet geometry

$$ds^2 = \frac{d\tau^2 + d\rho^2}{\cos^2\rho}, \qquad 0 \leq \rho \leq \frac{\pi}{2}, \quad \tau \sim \tau + b. \tag{18}$$

This geometry has an asymptotic boundary at $\rho = \frac{\pi}{2}$ at which we impose a Dirichlet boundary condition for the scalar field

$$\lim_{\rho \to \frac{\pi}{2}} \chi = +ik. \tag{19}$$

At the small end of the trumpet $\rho = 0$, we impose the following boundary conditions:

$$K = 0, \qquad \chi(\tau, 0) = j(\tau), \tag{20}$$

where $j(\tau)$ is a fixed function on the boundary circle. The condition $K = 0$ forces the small boundary to be the geodesic circle $\gamma$ at $\rho = 0$.

For simplicity, we will impose the additional condition

$$\int_\gamma d\tau \, \partial_\rho \Phi = 0, \tag{21}$$

which fixes the zero mode of $n^\mu \partial_\mu \Phi$ at the geodesic boundary $\gamma$. This avoids the need to introduce an additional boundary term to have a consistent variational problem. This zero

mode is not fixed by the JT equations of motion so it's possible to set it to zero.[5] The consistency of the variational problem with these boundary conditions is checked in Appendix C.1.

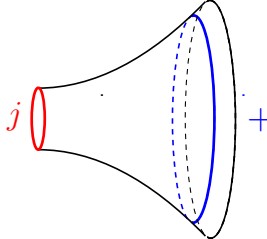

Figure 2: Half-wormhole saddle-point. The + represent the boundary source $+ik$. At the geodesic boundary, we impose a Dirichlet boundary condition on $\chi$.

We can think of these boundary conditions as corresponding to a *spacetime D-brane*, or SD-brane, which were discussed for example in [16, 49–52]. This terminology comes from the fact that if we view the string worldsheet as a two-dimensional spacetime, the string theory D-branes are SD-branes. The choice of $j(\tau)$ is understood as data specified by a specific SD-brane.

Although SD-branes are similar to end-of-the-world branes, considered for example in [18] for JT gravity, a conceptual difference is that they live in "superspace" whereas the end-of-the-world brane is associated to a single geometry. Hence, an SD-brane with boundary condition $j(\tau)$ allows any number of spacetimes to attach to it.

Specifying a Dirichlet boundary condition at the two boundaries $\rho = \frac{\pi}{2}$ and $\rho = 0$ uniquely fixes the solution to Laplace equation. So the half-wormhole is a classical solution of JT gravity with matter.

Our proposal will be that the choice of $j(\tau)$ corresponds to the choice of a single realization of the SYK couplings. Each realization of the couplings can be viewed as the introduction of an SD-brane with a particular choice for $j(\tau)$ and averaging over couplings corresponds to averaging over $j(\tau)$.

### 3.2 On-shell action

Let us now describe the solution in more detail and compute the on-shell action.

#### 3.2.1 Scalar action

The scalar field satisfies the Laplace equation $\Box\chi = 0$ together with the boundary condition

$$\chi(\tau, 0) = j(\tau), \qquad \lim_{\rho \to \pi/2} \chi(\tau, \rho) = \pm ik. \tag{22}$$

Let us focus for now on the $+ik$ choice, the other choice can be obtained by complex conjugation. We can use the Fourier decomposition

$$j(\tau) = \sum_{n \in \mathbb{Z}} j_n e^{2i\pi n\tau/b} \tag{23}$$

and reality of $j(\tau)$ imposes $j_n^* = j_{-n}$. In the case where we set to zero all the Fourier modes for $n \neq 1$, the solution is simply

$$\chi(\tau, \rho) = j_0 + \frac{2}{\pi}(ik - j_0)\rho. \tag{24}$$

---

[5]In the Euclidean wormhole [48], this zero mode would correspond to the asymmetry parameter $\eta$ which doesn't play an important role.

Including the higher Fourier modes, the profile of the scalar field can be parametrized from boundary sources by using bulk-to-boundary propagators. This allows to compute the scalar action and we obtain

$$S^{\chi} = -\frac{b}{\pi}(k + i j_0)^2 + 2 \sum_{n \geq 1} \frac{\pi n}{\tanh(\frac{\pi^2 n}{b})} |j_n|^2 . \tag{25}$$

The details of this derivation are given in the Appendix C.2.

We obtain the classical action of the half-wormhole by adding the JT contribution $b^2 T/2$ from the trumpet geometry [16]. This gives

$$S_{\text{half-WH}}^{\text{class.}} = \frac{b^2 T}{2} - \frac{b}{\pi}(k + i j_0)^2 + 2 \sum_{n \geq 1} \frac{\pi n}{\tanh(\frac{\pi^2 n}{b})} |j_n|^2 . \tag{26}$$

### 3.2.2 One-loop contribution

Let's now consider the one-loop piece. This piece turns out to be important. Since the theory with constant boundary sources is one-loop exact, this gives the exact partition function in the saddle-point approximation.

The one-loop piece coming from the scalar field can be obtained as follows. We conformally map the half-wormhole to a cylinder of width $\pi/2$ and circumference $b$. On the cylinder, the one-loop contribution is the thermal partition function

$$\text{Tr}\, e^{-2b\left(L_0 - \frac{1}{24}\right)} = \frac{1}{\eta(e^{-2b})} = \frac{e^{b/12}}{\prod_{n \geq 1}(1 - e^{-2nb})} , \tag{27}$$

in terms of the Dedekind eta function

$$\eta(q) = q^{1/24} \prod_{n \geq 1}(1 - q^n) . \tag{28}$$

In our large $b$ regime, we have $\prod_{n \geq 1}(1 - e^{-2nb}) = 1 + O(e^{-2b})$ which is negligible. The exponential then gives a Casimir contribution $-\frac{b}{12}$ to the action. There is also the term coming from the Weyl anomaly which is

$$\int_0^{\pi/2} d\rho \left(-\frac{b}{24\pi}\right) = \frac{b}{48} . \tag{29}$$

As a result, the classical action is corrected by the leading contribution of the Casimir energy of the scalar field

$$S_{\text{half-WH}}^{\text{Casimir}} = -\frac{b}{12} + \frac{b}{48} = -\frac{b}{16} . \tag{30}$$

We recall that in the wormhole, the leading Casimir contribution vanishes as the same computation gives [48]

$$S_{\text{WH}}^{\text{Casimir}} = -\frac{b}{24} + \frac{b}{24} = 0 . \tag{31}$$

Indeed, compared to the half-wormhole, the Casimir energy is divided by 2 and the Weyl anomaly is multiplied by 2.

Although it is a one-loop effect, the Casimir term cannot be neglected in the half-wormhole because it is of the same order as the other terms in the action. We will then write an effective action $S_{\text{half-WH}} = S_{\text{half-WH}}^{\text{class.}} + S_{\text{half-WH}}^{\text{Casimir}}$ which is explicitly

$$S_{\text{half-WH}} = \frac{b^2 T}{2} - \frac{b}{\pi}(k + i j_0)^2 + 2 \sum_{n \geq 1} \frac{\pi n}{\tanh(\frac{\pi^2 n}{b})} |j_n|^2 - \frac{b}{16} , \tag{32}$$

including the leading Casimir contribution. We will write the exact partition function as

$$Z_{\text{half-WH}} = z_{\text{half-WH}} \, e^{-S_{\text{half-WH}}}, \tag{33}$$

where the prefactor is

$$z_{\text{half-WH}} \equiv \sqrt{\frac{T}{2\pi}} \frac{1}{\prod_{n \geq 1}(1 - e^{-2nb})}, \tag{34}$$

contains the one-loop Schwarzian contribution and the remaining piece of the scalar field determinant. In our regime $\text{Re}\,b \gg 1$, this prefactor can be neglected.

### 3.2.3 Saddle-point in $b$

In the half-wormhole solution, the value of $b$, which measures the size of the geometry, is obtained by extremizing the action. This extremization should be done for the effective action (32) including the leading contribution from the Casimir energy.

The on-shell value $b = b_*$ is the solution of a transcendental equation which is easy to solve it numerically but doesn't admit a closed form.

We can obtain an analytical approximation under a mild assumption on $j_n$. Using $\tanh(x) \approx x$, we can approximate

$$\sum_{n \geq 1} \frac{\pi n}{\tanh(\frac{\pi^2 n}{b})} |j_n|^2 \approx \frac{b}{\pi} \sum_{n \geq 1} |j_n|^2. \tag{35}$$

This is valid for $b \gg 1$ and if $|j_n|^2$ decreases sufficiently fast with $n$. For example $|j_n| = O(n^{-4})$ would be sufficient. This gives

$$S_{\text{half-WH}} = \frac{b^2 T}{2} - \frac{b}{\pi}\left(\tilde{k}^2 - j_0^2 + 2ikj_0\right), \tag{36}$$

where we have introduced $\tilde{k}$ defined by

$$\tilde{k}^2 = k^2 - 2N(j) + \frac{\pi}{16}, \qquad N(j) \equiv \sum_{n \geq 1} |j_n|^2, \tag{37}$$

and we will assume that $\tilde{k} > 0$ as otherwise the saddle-point will be inconsistent. The extremization gives

$$b_* = \frac{1}{\pi T}(\tilde{k}^2 - j_0^2 + 2ikj_0). \tag{38}$$

This is a complex number so we see that the half-wormhole is a complex saddle-point. The real part is

$$\text{Re}\,b_* = \frac{1}{\pi T}\left(\tilde{k}^2 - j_0^2\right). \tag{39}$$

The partition function has a prefactor involving the Dedekind eta function (28) evaluated at $q = e^{-2b}$. The Dedekind eta function $\eta(q)$ is only defined on the region $|q| < 1$ and has singularities on the circle $|q| = 1$. As a result, the partition function only makes sense for

$$\text{Re}\,b > 0. \tag{40}$$

In the path integral, the contour of integration for $b$ is the semi-infinite line $[0, +\infty)$. If there is a saddle-point in the region $\text{Re}\,b > 0$, we can deform the contour to pass through this point and use the saddle-point approximation. However, this is not possible for a saddle-point in the

region Re $b < 0$ because the contour cannot be deformed past the line Re $b = 0$. Hence, saddle-points in the region Re $b < 0$ should not contribute. The conclusion is that the half-wormhole should only be included if

$$|j_0| < \tilde{k}. \tag{41}$$

It is easy to check that our regime Re $b \gg 1$ is valid close to the critical temperature using that $S_0 \gg 1$. The effective action is then

$$S_{\text{half-WH}} = -\frac{1}{2\pi^2 T}(\tilde{k}^2 - j_0^2 + 2ikj_0)^2 = -\frac{T}{2}b_*^2. \tag{42}$$

### 3.3 Half-wormhole at $k = 0$

It is interesting to note that the half-wormhole solution persists at $k = 0$, *i.e.* without having any boundary source for the scalar field. Indeed, we have Re $b_* > 0$ for $k = 0$ if $j(\tau)$ satisfies

$$j_0^2 + 2N(j) < \frac{\pi}{16}. \tag{43}$$

For example, we could simply take $j(\tau) = 0$. In such a regime, the half-wormhole saddle-point has to be included. The possibility of such a solution is a consequence of the imperfect cancellation of the leading Casimir energy in the half-wormhole as discussed in section 3.2.2. This leaves a large negative energy which makes such a solution possible. In the wormhole, the leading Casimir energy cancels and there is no solution at $k = 0$.

As the parameter $k$ is dual to the parameter $\kappa$ in SYK, this solution might be related to the standard SYK model (with real couplings). In particular, we have noticed that the $k = 0$ half-wormhole gives a "noisy" contribution to the spectral form factor. However, it doesn't produce a ramp as one would have hoped so perhaps additional saddle-points are needed to understand a single realization of the spectral form factor. In this paper, we focus on finite $k$ because we want the wormhole solution to be present. It would be interesting to understand better the meaning of the $k = 0$ half-wormhole.

## 4 Gravity without average

In this section, we study the "exact" gravity theory corresponding to a fixed choice of $j(\tau)$. In this theory, we don't include wormholes and factorization is manifest. We will show in The theory obtained by averaging over $j(\tau)$ is discussed in the next section. The interpretation of this theory as the non-averaged gravity theory will be verified in section 6 by comparing it with a single realization of the SYK model.

**One boundary.** Let's first consider the problem with one boundary and with a source $+ik$ for the scalar field. There are two contributions: the disk and the half-wormhole:

$$Z_+(\beta) = Z^{\text{disk}}(\beta) + Z_+^{\text{half-WH}}(\beta). \tag{44}$$

This is illustrated as

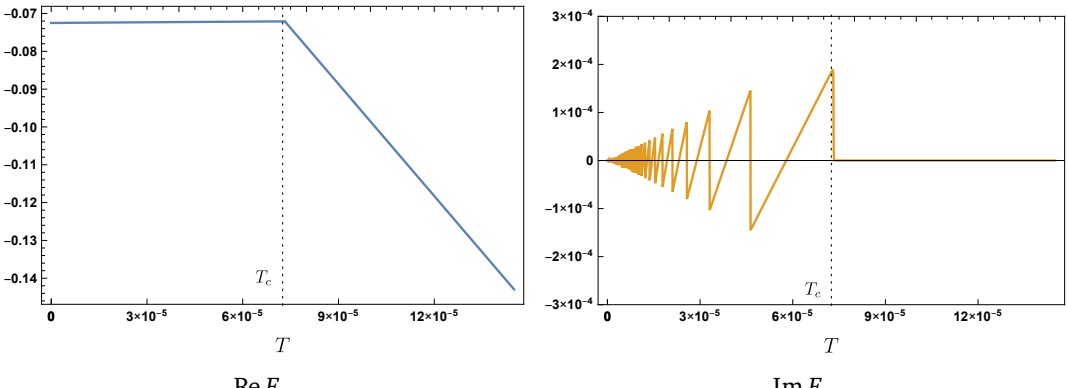

Figure 3: Real and imaginary parts of the free energy as a function of the temperature. We observe a distinctive saw-like pattern in the imaginary part which will also appear in SYK. The parameters used for these plots are $k = 1, S_0 = 10^3, N(j) = 0, j_0 = 3 \times 10^{-3}$.

The disk partition function is given by

$$Z^{\text{disk}}(\beta) = z_{\text{BH}} \, e^{S_0 + 2\pi^2 T} \,, \qquad z_{\text{BH}} \equiv \frac{T^{3/2}}{\sqrt{2\pi}} \,, \tag{45}$$

and the prefactor can be neglected in our regime.

Including both saddle-points, the total free energy is

$$
\begin{aligned}
F &= -T \log\left[ Z_+^{\text{disk}} + Z_+^{\text{half-WH}} \right] \\
&= -T \log\left[ z_{\text{BH}} \exp\left( S_0 + 2\pi^2 T \right) + z_{\text{half-WH}} \exp\left( \frac{1}{2\pi^2 T} (\tilde{k}^2 - j_0^2 + 2ikj_0)^2 \right) \right].
\end{aligned}
\tag{46}
$$

The prefactor will never be important in our regime and we might as well set $z_{\text{BH}} = z_{\text{half-WH}} = 1$. We can compute the critical temperature from the equation

$$S_{\text{BH}} = \operatorname{Re} S_{\text{half-WH}} \,. \tag{47}$$

This gives

$$T_c = \frac{1}{2\pi^2 S_0} \left( (\tilde{k}^2 + j_0^2)^2 - 4k^2 j_0^2 \right) + O(S_0^{-2}), \tag{48}$$

where we have used $S_0 \gg 1$. We see that $T_c$ is positive only in the region[6]

$$|j_0| < j_0^* = \sqrt{k^2 + \tilde{k}^2} - k \,. \tag{49}$$

Above this point, only the black hole dominates.

For $|j_0| < j_0^*$, the free energy is as in Fig. 3. We see the phase transition at low temperature. Below the critical temperature, the real part of the free energy is flat and the imaginary part has a distinctive saw-like structure. Above the critical temperature, the free energy is that of the black hole.

For $|j_0| > j_0^*$, we only have the black hole. For $k = 1$ and $N(j) = 0$, the transition is at $j_0^* \approx 0.482$.

In the range where it is small, the parameter $j_0$ can be seen to be related to the number of oscillations in the imaginary part. To illustrate this, we have plotted in Fig. 4 the imaginary part with the same parameters as in Fig. 3 but with different values of $j_0$.

---

[6]It becomes positive again for larger $|j_0|$ but this is beyond (41) so the half-wormhole should not be included there.

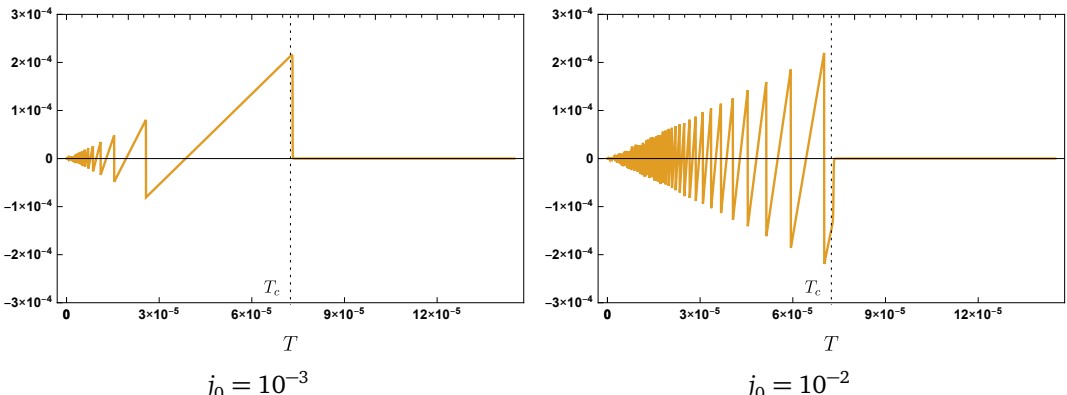

$$j_0 = 10^{-3} \qquad\qquad\qquad j_0 = 10^{-2}$$

Figure 4: Imaginary part of the free energy as a function of the temperature. We see that $j_0$ is related to the number of oscillations. A similar behavior will be observed in SYK for the dual parameter $r$. The parameters used for these plots are $k = 1, S_0 = 10^3, N(j) = 0$.

We can understand the saw-like pattern by computing

$$\mathrm{Im}\, F = -T \arg\left[ Z_+^{\mathrm{disk}} + Z_+^{\mathrm{half\text{-}WH}} \right], \tag{50}$$

where arg is the complex argument function valued in $[-\pi, \pi]$. Below the critical temperature, the black hole can be neglected and we have

$$\mathrm{Im}\, F \approx -T \arctan\left( \tan\left( \frac{2k j_0 (\tilde{k}^2 - j_0^2)}{\pi^2 T} \right) \right), \tag{51}$$

where the function $\arctan(\tan(x))$ is just used to give the modulo $2\pi$ representative of $x$ in $[-\pi, \pi]$. This formula precisely reproduces the saw-like pattern. As described in section 6, this pattern is also observed in a single realization of the SYK model.

**Two boundaries.** Let's now consider two boundaries with $\pm ik$. We are computing

$$Z_-(\beta)Z_+(\beta) \;=\; \left( \; - \!\!\bigcirc\!\! \; + \; - \!\!\bigcirc\!\!j \; \right) \times \left( \; \bigcirc\!\!+ \; + \; j\bigcirc\!\!+ \; \right)$$

$$= \; - \!\!\bigcirc\!\bigcirc\!\!+ \; + \; - \!\!\bigcirc\!j\bigcirc\!\!+ \; + \; - \!\!\bigcirc\!j\bigcirc\!\!+ \; + \; - \!\!\bigcirc\!j\bigcirc\!\!+$$

The answer manifestly factorizes since we don't include wormholes. The free energy is simply

$$F = -T \log(Z_- Z_+) = 2\,\mathrm{Re}(-T \log Z_-), \tag{52}$$

which is twice the real part of the one boundary free energy. The analysis of the previous section applies. We have a phase transition from a phase with two black holes to a phase with two half-wormholes. The imaginary part vanishes here.

Although the Euclidean wormhole has been excluded by hand here, there is a sense in which it is still present in the contribution of the two half-wormholes. This can be seen in the fact that the free energy here is qualitatively similar to the free energy of the theory with only black holes and wormholes, reviewed in section 2.

Indeed, it is natural to isolate the self-averaging part from the contribution of the two half-wormholes. We will show in the next section that this gives precisely the wormhole. We can then view the two half-wormholes as the sum of the wormhole contribution and a non self-averaging piece, represented by a linked contribution in [35].

## 5 Averaged gravity theory

In this section, we will consider the effect of the average over $j(\tau)$. We will consider the following theories:

- a "simple" gravity theory, here JT gravity with a massless scalar, defined by the Euclidean path integral including wormhole geometries. This theory doesn't factorize.

- an "exact" gravity theory, defined from the simple gravity by removing wormholes and allowing half-wormholes with boundary condition $j(\tau)$. This theory manifestly factorizes.

The main result of this section will be that simple gravity is precisely equivalent to the average over $j(\tau)$ of the exact theory for a suitable choice of ensemble.

### 5.1 Ensemble of boundary conditions

In general, we can consider an ensemble average of the form

$$\langle \mathcal{O}[j(\tau)]\rangle = \mathcal{N} \int Dj(\tau)\, e^{-\mathcal{S}[j(\tau)]}\mathcal{O}[j(\tau)], \tag{53}$$

for any quantity $\mathcal{O}$. In Fourier space, we have the decomposition (23). The integration measure will be taken to be

$$Dj(\tau) = \prod_{n\in\mathbb{Z}} |dj_n| = \prod_{n\in\mathbb{Z}} (2dj_n^{(R)} dj_n^{(I)}), \tag{54}$$

writing $j_n = j_n^{(R)} + i j_n^{(I)}$. $\mathcal{N}$ is an appropriate normalization factor. There are many possible choices for $\mathcal{S}[j(\tau)]$. It is natural to make a choice that depends only on the Fourier modes, assigning less weight to the higher modes. For example, we can take:

$$\text{first ensemble} \quad : \quad \mathcal{S}[j(\tau)] = \frac{1}{2J^2}\sum_{n\in\mathbb{Z}}(1+|n|)|j_n|^2, \tag{55}$$

$$\text{second ensemble} \quad : \quad \mathcal{S}[j(\tau)] = \frac{1}{2J^2}\sum_{n\in\mathbb{Z}}(1+n^4)|j_n|^2. \tag{56}$$

A typical realization of $j(\tau)$ in these two ensembles is represented in Fig. 5. This illustrates the fact that a single realization of $j(\tau)$ can contain a lot of complexity. For any such realization, there will be a half-wormhole saddle-point. The on-shell action (25) is sensitive to each Fourier mode so the half-wormhole will be very much dependent on the specific realization. The on-shell value $b$ will also depend on all the Fourier modes. In this respect, the half-wormhole is not a self-averaging object.

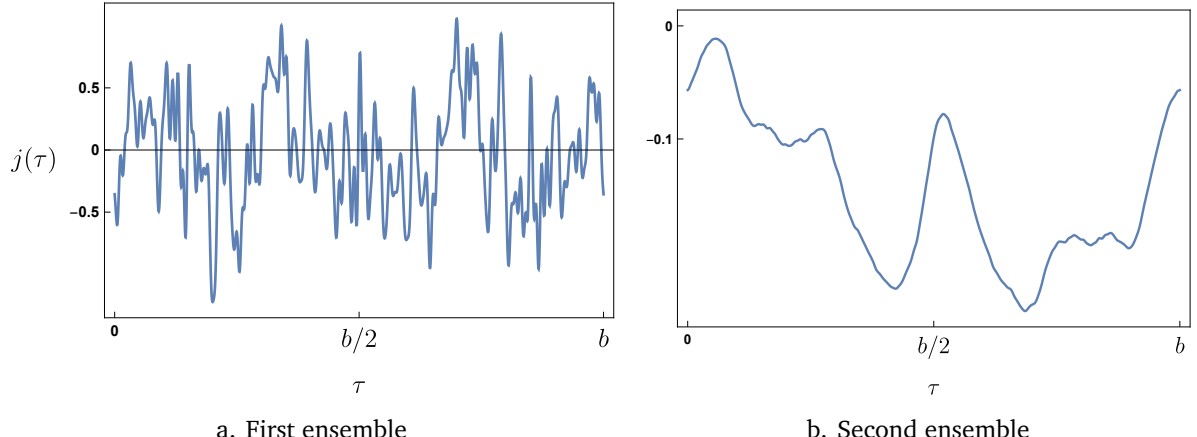

| a. First ensemble | b. Second ensemble |

Figure 5: Typical realization of $j(\tau)$.

In the above ensemble, the typical value of $j_n$ is

$$\text{first ensemble} \quad : \quad \langle j_n^2 \rangle \sim \frac{J^2}{1+|n|}, \tag{57}$$

$$\text{second ensemble} \quad : \quad \langle j_n^2 \rangle \sim \frac{J^2}{1+n^4}. \tag{58}$$

A limit that will be important is the "uniform" limit $J \to +\infty$. We will see that this is the limit where the average of the exact theory corresponds to the effective theory. As this limit is singular, some rescaling will be necessary.

We will define non-normalized averages as

$$\langle \mathcal{O}[j(\tau)] \rangle_0 = \int Dj(\tau)\, e^{-\mathcal{S}[j(\tau)]} \mathcal{O}[j(\tau)], \tag{59}$$

and the true average is

$$\langle \mathcal{O}[j(\tau)] \rangle = \frac{\langle \mathcal{O}[j(\tau)] \rangle_0}{\langle 1 \rangle_0}. \tag{60}$$

The difference between normalized and non-normalized average won't be too important. If we average the partition function, this corresponds to a term $\log \langle 1 \rangle_0 \sim \log J$ in the action which should not play an important role. For example, in the first ensemble, we have

$$\langle 1 \rangle_0 = \prod_{n \geq 1} \frac{4\pi J^2}{1+|n|} = 4\pi J^2 \prod_{n \geq 1} \frac{4\pi J^2}{n} = \sqrt{2}J, \tag{61}$$

using a standard zeta regularization for the infinite product [59]. It will be possible to make the limit $J \to +\infty$ regular by adding appropriate "weight factors" in the averaging procedure, to take care of these $J$-dependent factors.

## 5.2 Simple gravity as an average

In this section, we explain how simple gravity emerges from the average over $j(\tau)$ in the limit $J \to +\infty$.

### 5.2.1 Two boundaries: emergence of the wormhole

Let's first consider the averaged two-boundary problem. We have the four contributions depicted in section 4 and the average gives

$$\langle Z_- Z_+ \rangle = Z_-^{\text{BH}} Z_+^{\text{BH}} + \langle Z_-^{\text{half-WH}} Z_+^{\text{half-WH}} \rangle, \tag{62}$$

where we used that the average over single half-wormholes is negligible as will be shown in the next section. The failure of factorization is given by

$$\langle Z_- Z_+ \rangle - \langle Z_- \rangle \langle Z_+ \rangle = \langle Z_-^{\text{half-WH}} Z_+^{\text{half-WH}} \rangle. \tag{63}$$

This is given by

$$Z_-^{\text{half-WH}} Z_+^{\text{half-WH}} = z_{\text{hWH}}^2 \, e^{-S_{\text{pair}}}, \tag{64}$$

where $S_{\text{pair}}$ is twice the real part of the half-wormhole effective action

$$S_{\text{pair}} = b^2 T - \frac{2b}{\pi} k^2 + \frac{2b}{\pi} j_0^2 + 4 \sum_{n \geq 1} \frac{\pi n}{\tanh(\frac{\pi^2 n}{b})} |j_n|^2 - \frac{b}{8}, \tag{65}$$

and the prefactor is the square of (34).

We will perform the average before extremizing over $b$. This is possible because we are computing a double integral over $j(\tau)$ and $b$ (approximating the integral over $b$ by a saddle-point) and the order of the integrals doesn't matter.

We consider the $J \to +\infty$ limit of the average where the measure becomes simpler. In the unnormalized average, we have the contribution

$$\int d j_0 \exp\left(-\frac{2b}{\pi} j_0^2\right) \prod_{n \geq 1} \int |d j_n \, d j_{-n}| \exp\left(-\frac{4\pi n}{\tanh(\frac{\pi^2 n}{b})} |j_n|^2\right) = \frac{\pi}{\sqrt{2b}} \prod_{n \geq 1} \frac{1}{2n} \tanh\left(\frac{\pi^2 n}{b}\right). \tag{66}$$

So this gives a multiplicative factor to the partition function. We can use the identity

$$\prod_{n \geq 1} \tanh\left(\frac{\pi^2 n}{b}\right) = e^{-b/8} \sqrt{\frac{2b}{\pi}} \prod_{n \geq 1} \frac{(1 - e^{-2nb})^2}{1 - e^{-nb}}. \tag{67}$$

This expression can be obtained by writing the product in terms of the Dedekind eta function and using its modular transformation, see Appendix C.3.

The product gives an exponentially suppressed contribution at large $b$ but there is a non-trivial "Casimir energy" $e^{-b/8}$, which should be included in the action. In fact, we see that this piece precisely cancel the $-b/8$ already present in the action due to the non-zero Casimir energy of the half-wormholes.

We can use that the expression (67) and the fact that $\prod_{n \geq 1}(2n) = \sqrt{\pi}$ from zeta regularization [59] to show that the result is exactly the wormhole partition function:

$$\lim_{J \to +\infty} \langle Z_-^{\text{half-WH}} Z_+^{\text{half-WH}} \rangle_0 = Z_{\text{WH}} = z_{\text{WH}} \, e^{-S_{\text{WH}}} \tag{68}$$

given by

$$S_{\text{WH}} = b^2 T - \frac{2b}{\pi} k^2, \qquad z_{\text{WH}} = \frac{T}{2\pi} \frac{1}{\prod_{n \geq 1}(1 - q^n)}. \tag{69}$$

Remarkably, we obtain the correct one-loop prefactor. Finally, the saddle-point over $b$ gives $b = k^2/(\pi T)$ and a constant free energy, as reviewed in section 2.1. This shows that the wormhole precisely arises from an average over half-wormholes. Note that this fact does not

rely on a particular choice of ensemble for $j(\tau)$ but only requires the $J \to +\infty$ limit where the measure becomes uniform.

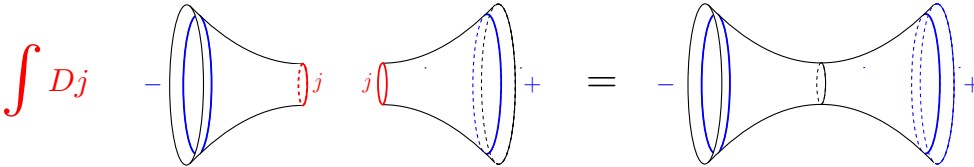

Figure 6: Emergence of the wormhole from the average over boundary conditions.

There is a simple way to understand why the wormhole partition function is exactly equal to the average of two half-wormholes in the $J \to +\infty$ limit. The half-wormhole boundary condition fixes the value of the scalar field to be $j(\tau)$ at the geodesic boundary. If we consider two half-wormholes with the same $j(\tau)$, we can glue them and obtain a wormhole with a non-trivial profile for the scalar field. Integrating over $j(\tau)$ simply corresponds to "finishing" the path integral. In this way, the average comes from the fact that we isolate a specific field in the path integral, and we decide to integrate over it later.

Note that this was for the unnormalized average. The true average is given by

$$\langle Z_-^{\text{half-WH}} Z_+^{\text{half-WH}} \rangle = \frac{\langle Z_-^{\text{half-WH}} Z_+^{\text{half-WH}} \rangle_0}{\langle 1 \rangle_0} . \tag{70}$$

At large $J \gg 1$, we obtain a correction to the wormhole action of order $\log \langle 1 \rangle_0 \sim \log J$ which can be consider small if $J$ is not too large.

To obtain a finite limit $J \to +\infty$, we should rescale the half-wormhole partition function and define

$$\widetilde{Z}_-^{\text{half-WH}} = \langle 1 \rangle_0^{1/2} Z_-^{\text{half-WH}} , \tag{71}$$

so that

$$\langle \widetilde{Z}_-^{\text{half-WH}} \widetilde{Z}_+^{\text{half-WH}} \rangle = \langle Z_-^{\text{half-WH}} Z_+^{\text{half-WH}} \rangle_0 = Z_{\text{WH}} , \tag{72}$$

gives correctly the wormhole. This can be seen as the addition of a constant term to the action of the half-wormhole. This can be viewed as part of the averaging procedure. The important point here is that we have obtained the precise relation between the wormhole in simple gravity and an average of the half-wormholes of the exact theory.

### 5.2.2 One boundary: disappearance of the half-wormhole

For a single boundary, we have

$$\langle Z_- \rangle = Z_-^{\text{BH}} + \langle Z_-^{\text{half-WH}} \rangle . \tag{73}$$

In this limit $J \to +\infty$, we have

$$\lim_{J \to +\infty} \langle Z^{\text{hWH}} \rangle_0 = z^{\text{hWH}} \int Dj(\tau) \exp\left[ -\frac{b^2 T}{2} + \frac{b}{\pi}(k + ij_0)^2 - 2\sum_{n \geq 1} \frac{\pi n}{\tanh(\frac{\pi^2 n}{b})} |j_n|^2 + \frac{b}{16} \right] , \tag{74}$$

where we recall that the measure is $Dj(\tau) = \prod_{n \in \mathbb{Z}} |dj_n|$.

Firstly, we can see that the integral over $j_0$ removes the dependence on $k$, as can be seen by doing a simple change of variable. This is one of the reason the half-wormhole saddle-point will not survive the average. This integral generates the prefactor $\pi/\sqrt{b}$.

We then perform the integral over $j_n$ for $n \neq 0$. This gives the contribution

$$\prod_{n \geq 1} \int |dj_n dj_{-n}| \exp\left(-\frac{2\pi n}{\tanh(\frac{\pi^2 n}{b})}|j_n|^2\right) = \prod_{n \geq 1} \frac{1}{n}\tanh\left(\frac{\pi^2 n}{b}\right).$$

Again, this contribution cannot be neglected in a saddle-point analysis because of the $e^{-b/8}$ term in (67).

Combining everything, we obtain

$$\lim_{J \to +\infty} \left\langle Z^{\text{hWH}} \right\rangle_0 = z_{\text{av}}^{\text{hWH}} \exp\left(-S_{\text{av}}^{\text{hWH}}\right), \tag{75}$$

with the effective action and prefactor being

$$S_{\text{av}}^{\text{hWH}} = \frac{b^2 T}{2} + \frac{b}{16}, \qquad z_{\text{av}}^{\text{hWH}} = \sqrt{\frac{T}{2\pi}}\prod_{n \geq 1}(1 + e^{-nb}). \tag{76}$$

There is saddle-point at $b_* = -\frac{1}{16T}$ but because this is negative, this saddle-point should not be included as explained in section 3.2.3. The half-wormhole saddle-point has disappeared in the average.

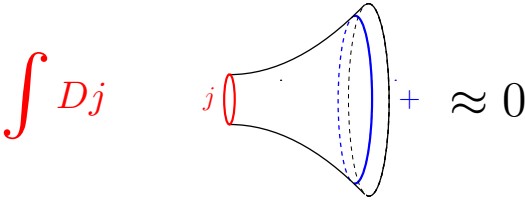

Figure 7: Disappearance of the half-wormhole saddle-point in the average.

There were two important effects that are in play here. Firstly, the average over the zero mode $j_0$ has removed the dependence on the source $k$. Then, the average over $j_n$ has changed the sign of the Casimir energy and prevented the existence of a saddle-point at positive $b$.

In the averaging procedure described above, we obtain the true average as

$$\left\langle \widetilde{Z}_-^{\text{half-WH}} \right\rangle = \langle 1 \rangle_0^{-1/2} \langle Z_-^{\text{half-WH}} \rangle_0, \tag{77}$$

which is still suppressed.

As a result, the average of the one boundary partition function is dominated by the black hole

$$\langle Z_- \rangle \approx Z_-^{\text{BH}}. \tag{78}$$

This precisely reproduces the expectation from simple gravity.

### 5.2.3 Quenched average

The quenched average for a single boundary is obtained by averaging the free energy of a single realization discussed in section 4. It's easy to see that the real part, which is self-averaging, remains of the same form, but with a smoother transition. The imaginary part averages to zero. An example with 100 realizations of $j(\tau)$ is given in Figure 8.

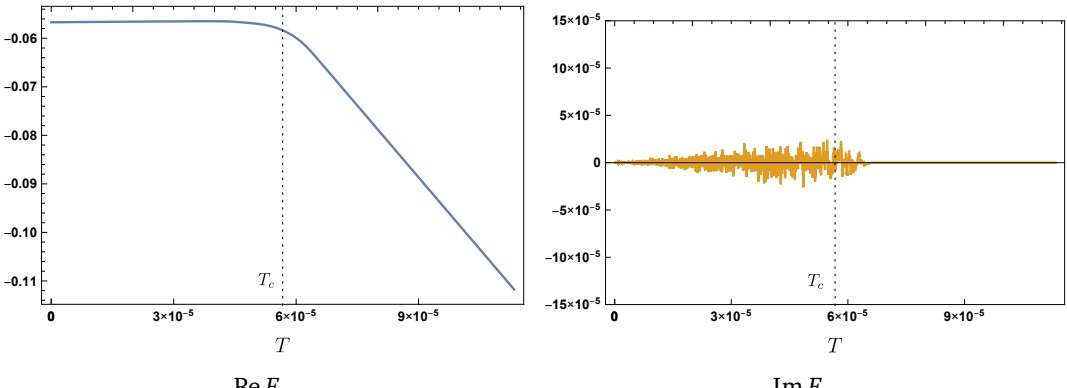

Figure 8: Quenched average over 100 realization of $j(\tau)$ in the first ensemble (55). The parameters used for the plot are $k = 1, S_0 = 10^3, J = 0.1$. The vertical scale for the imaginary part is adapted for one realization so that we see that it averages to zero. $T_c$ is defined as the mean of the critical temperatures of the different realizations.

If we have two boundaries, we can use that

$$\langle \log(Z_+ Z_-) \rangle = 2 \operatorname{Re} \langle \log Z_+ \rangle, \tag{79}$$

so that the quenched free energy is twice the real part of the one boundary quenched free energy, and the discussion is the same as above. This corresponds to our wormhole solution.

How does this compare to simple gravity? It seems that for one boundary, simple gravity has only the black hole and hence cannot have the flat part seen in the quenched average of the exact theory. However, this is too naive. To compute the real part of the quenched free energy in gravity, we actually need to introduce the complex conjugate and write

$$\operatorname{Re} \langle \log Z_+ \rangle = \frac{1}{2} \langle \log(Z_+ Z_-) \rangle. \tag{80}$$

This allows for the possibility of a wormhole contribution connecting the boundary to its complex conjugate, a "real part wormhole". This is precisely what happens here and reproduces the flat part in the free energy. This shows that the quenched free energy of the exact theory is also reproduced by simple gravity.

Finally, we have shown that the average over $j(\tau)$ in the limit $J \to +\infty$ is precisely equivalent to simple gravity, for both the annealed and quenched averages.

## 5.3  More general averages

In the previous section, we considered the average in the limit $J \to +\infty$. This is the regime in which the average theory becomes the simple gravity theory. In this section, we will discuss more general possibilities.

For simplicity, we will freeze $j_n$ for $n \neq 1$ and consider the average over only $j_0$. Also, we we will use the approximation described in section 3.2.3 which requires that $|j_n|$ decreases sufficiently fast with $n$. The values of $j_n, n \neq 1$ are then completely captured in the on-shell action by the constant $N(j)$ defined in (37).

The average of the partition function involves an integral over $j_0$ that is difficult to analyze. A simpler method is to do the average before the saddle-point over $b$. We thus consider

$$\langle Z_+(\beta) \rangle = \mathcal{N} \int D j_0 \, e^{-j_0^2/(2J^2)} e^{-S_{\text{half-WH}}}, \tag{81}$$

using the expression (36). This gives

$$\langle Z_+(\beta)\rangle = \mathcal{N}' e^{-S_{\text{annealed}}}, \qquad S_{\text{annealed}} = \frac{b^2 T}{2} + \frac{b}{\pi}\left(\frac{k^2}{1+\frac{\pi}{2J^2 b}} - \tilde{k}^2\right), \tag{82}$$

with unimportant normalization constants $\mathcal{N}$ and $\mathcal{N}'$.

The integral over $b$ remains to be done. It can be estimated by a saddle-point approximation. The saddle-point equation

$$\frac{dS_{\text{annealed}}}{db} = 0, \tag{83}$$

is a cubic equation in $b$ which leads to three saddle-points. Here, we will simplify the problem and only discuss the regimes $bJ^2 \ll 1$ and $bJ^2 \gg 1$.

**Regime $bJ^2 \ll 1$.** This is compatible with $b \gg 1$ only for $J \ll 1$. In this regime, we can neglect the 1 in the denominator. We find a saddle-point at

$$b_* = \frac{\pi \tilde{k}^2}{\pi^2 T + 4J^2 k^2}, \tag{84}$$

which is always positive. The validity of the above approximation requires $b_* J^2 \ll 1$. Since we keep $k$ and $\tilde{k}$ of order one, this is only possible if the temperature term dominates. The critical temperature is of the order

$$T_c \sim \frac{\tilde{k}^4}{2\pi^2 S_0}, \tag{85}$$

neglecting the $j_0$ term since $J \ll 1$. This gives the condition

$$J \ll S_0^{-1/2}, \tag{86}$$

and we also have $S_0 \gg 1$ so this corresponds to very small values of $J$. All the approximations are then valid. The resulting on-shell action is

$$S_{\text{annealed}} = -\frac{\tilde{k}^4}{2\pi^2 T}, \qquad (bJ^2 \ll 1). \tag{87}$$

**Regime $bJ^2 \gg 1$.** Since we already have $b \gg 1$, this corresponds to $J \gtrsim 1$. In this regime, we have

$$S_{\text{annealed}} \approx \frac{b^2 T}{2} + \frac{b}{\pi}\left(k^2 - \tilde{k}^2\right) \qquad (bJ^2 \gg 1), \tag{88}$$

which gives a saddle-point at

$$b_* = \frac{\tilde{k}^2 - k^2}{\pi T} = \frac{1}{\pi T}\left(\frac{\pi}{16} - 2N(j)\right). \tag{89}$$

Note that this is a real saddle-point. It should only be included for $b_* > 0$. The resulting on-shell action is

$$S_{\text{annealed}} = -\frac{(\tilde{k}^2 - k^2)^2}{2\pi^2 T}, \qquad (bJ^2 \gg 1). \tag{90}$$

We should check that $b_* \gg 1$ for our approximations to be valid. The critical temperature corresponds to $S_{\text{annealed}} = -S_0$ which leads to $T_c \sim (\tilde{k}^2 - k^2)^2/S_0$. Close to the critical temperature, we then have

$$b_* \sim \frac{S_0}{\pi(\tilde{k}^2 - k^2)}, \tag{91}$$

so we have $b_* \gg 1$ as long as $\tilde{k}^2 - k^2$ stays of order one.

This also captures the limit $J \to +\infty$. As the half-wormhole should only be included for $b_* > 0$, there are two possible cases

1. If $\tilde{k} > k$, i.e. $N(j) < \frac{\pi}{32}$, the half-wormhole is still present in the $J \to +\infty$ limit,

2. If $\tilde{k} < k$, i.e. $N(j) > \frac{\pi}{32}$, the half-wormhole disappears in the $J \to +\infty$ limit.

So we see that depending on the choice of parameters, the half-wormhole either survives or disappears in the $J \to +\infty$ limit. A natural choice is to set $j_n = 0$ for $n \geq 1$ which corresponds to $N(j) = 0$. This leads to the first case and the half-wormhole survives in the $J \to +\infty$ limit. Instead, we can perform the average over the $j_n$ as in the previous section. As showed there, this leads to an effective value $N(j) = \frac{\pi}{16}$ because of the identity (67). We are then in the second case and the half-wormhole disappears.

## 6 SYK without average

We have described above a gravity theory in which half-wormhole saddle-points are important. We have also shown how the wormhole emerges after ensemble average over SD-brane boundary conditions. Following the close relation between this wormhole and a two-site SYK model with complex couplings [48], reviewed in section 2, we naturally expect that the field theory associated to the gravity theory before average, termed *exact*, is a single realization of the one-site SYK model with Hamiltonian

$$ H \;=\; \frac{1}{4!} \sum_{i,j,k,\ell=1}^{N/2} (J_{ijk\ell} + i\kappa M_{ijk\ell}) \psi_i \psi_j \psi_k \psi_\ell . \tag{92} $$

In this section, we confirm this expectation by computing the free energy of a single disorder realization of a SYK model with complex couplings and comparing it with the free energy of the exact gravity theory.

We will not perform any average but just study how the free energy depends on the couplings. As we do not want to study all the complicated features of a single realization, we will focus on the dependence on the mean value

$$ z = \overline{J_{ijk\ell} + i\kappa M_{ijk\ell}} \tag{93} $$

defined by summing over $i, j, k, \ell$ and dividing by the number of terms. To any single realization of the complex couplings, we can associate this complex number $z$. This quantity has the interpretation of a coupling-dependent ground state energy. For example, a simple way to change the value of $z$ is to shift the complex couplings by a fixed constant. Overall, this just adds a constant to the Hamiltonian after using anticommutation relations. Our purpose in isolating this simple parameter is to make the comparison with the gravity side and we will see that $z$ appears to be closely related with the zero mode $j_0$ of $j(\tau)$.

We take $J_{ijk\ell}$ and $M_{ijk\ell}$ to be Gaussian variable with zero mean and standard deviation $\sqrt{\langle J_{ijk\ell}^2 \rangle} = \sqrt{\langle M_{ijk\ell}^2 \rangle} = (12/N)^{3/2}$. For a typical realization in this ensemble, the parameter $z$ will be very small due to the large amount of cancellations. For instance, we typically have $|z| \sim 10^{-4}$ for $N = 34$. However, we stress we are not considering typical realizations. We are merely studying how a function (the free energy) depends on its variables (the complex couplings). For this purpose, we are not restricted to typical realizations and it will be convenient to consider realizations where $z$ has a much larger value.

To sample a realization with a tunable $z$, a convenient procedure is to take the $J_{ijk\ell}$ and $M_{ijk\ell}$ mentioned above and add some fixed complex value $z$ to the complex couplings. This gives a single realization where the mean value is approximately $z$ as long as $z$ is not too small so that we can neglect the Gaussian noise in the center-of-mass. We will write $z = re^{i\theta}$.

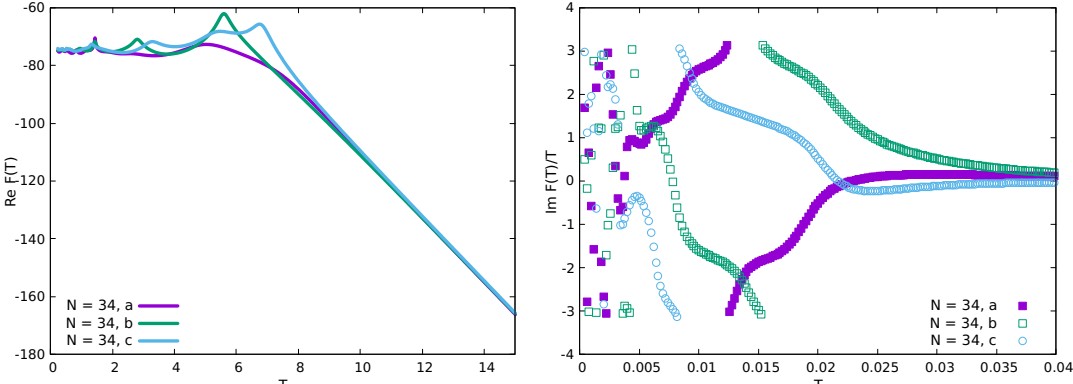

Figure 9: Left: Real part of the free energy for a single disorder realization of the Hamiltonian (92) for $N = 34$ and $z = re^{i\theta} = 0$. We consider a few disorder realizations in order to illustrate the dependence of the results on the choice of random couplings. This dependence is relatively minor. In all cases, we do observe clearly a flat part for low temperature and a rather abrupt decreases at a finite temperature $T_c$ consistent with a first order phase transition. This is similar to both the prediction of gravity of the previous section and also the free energy after ensemble average of the two-site complex SYK [48] related to a Euclidean wormhole. Right: Imaginary part of the free energy for the same parameters. Results are rather sensitive to the details of the disorder realization.

The eigenvalues of the Hamiltonian (92) are obtained numerically by exact diagonalization techniques. The gravity results assume a large $N$ limit so we will focus on the largest $N = 34$ that we can reach numerically. Unless stated otherwise, we set $\kappa = 1$. For convenience, so that the relevant values of $r$ are of order one, we redefine $r \to r/N$. A more natural scaling [60] may be $r \to r/N^{3/2}$. However, we stress again that since we do not take any ensemble average, we are free to choose the details of the random couplings.

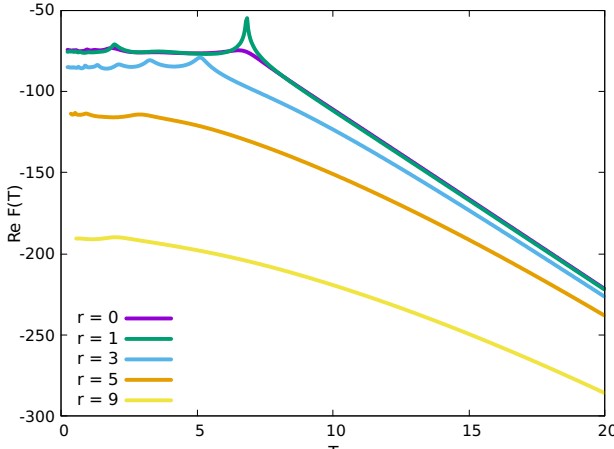

Figure 10: Real part of the free energy for $z = re^{i\pi/6}$ for $N = 34$ and different $r$. As $r$ increases, we observe a gradual reduction of the low temperature flat part until it almost disappears for $r > 5$. This feature is consistent with the gravity result of the previous section.

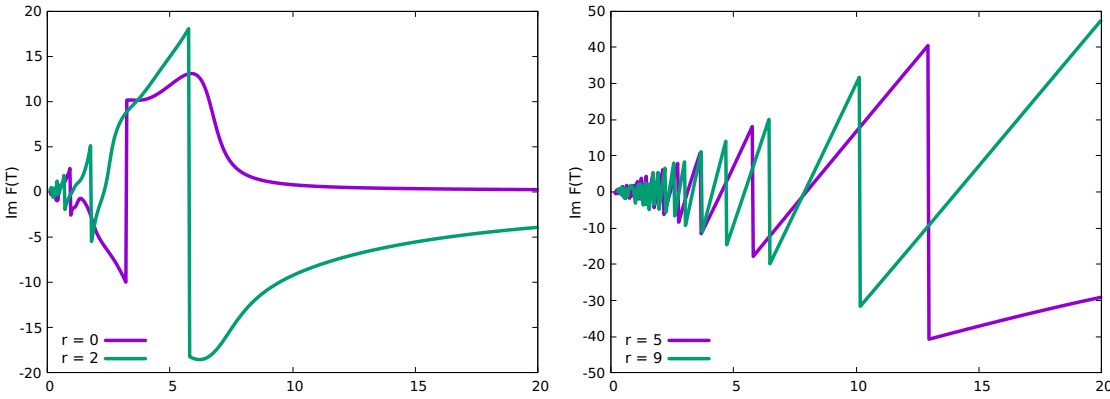

Figure 11: Imaginary part of the free energy for $z = re^{i\pi/6}$, $N = 34$ and different $r$'s. As $r$ increases, the free energy develops a characteristic saw-like shape.

After these considerations, the free energy

$$F = -T \log Z ,\tag{94}$$

is computed by an explicit evaluation of the partition function. Note that since the spectrum is complex, the free energy will have in general real and imaginary parts. As far as we are aware, not much is known about the imaginary part of the free energy in the context of the SYK model. The first question we would like to clarify is whether a single disorder realization of this SYK model captures the main features of the disorder average case, reviewed in section 2.2.

Results depicted in Fig. 9 confirm that this is largely the case. Despite the natural increase of fluctuations, we still observe a flat part at low temperature that ends rather abruptly at a critical temperature $T_c$. Although details depend on the disorder realization, these general features are rather robust.

We now explore the effect of a finite $z$. We first fix $\theta = \pi/3$ and change $r$. For small $r$, see Fig. 10, we do not observe great differences in the real part of the free energy with respect to the $r = 0$ case. However, as $r$ increases, the flat part is gradually reduced and ultimately, for sufficiently large $r$, seems to completely vanish. This is in qualitative agreement with the

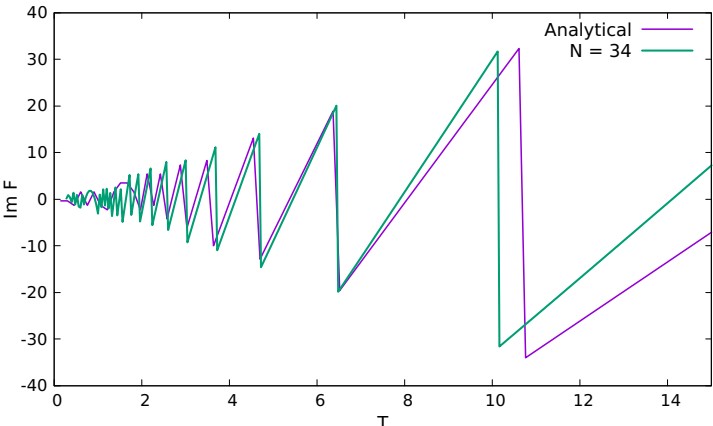

Figure 12: Imaginary part of the free energy for $z = 9e^{i\pi/6}$. We find an excellent agreement with the gravity prediction Eq. (51) with fitting parameters that provide an effective relation between the SYK and the gravity system parameters.

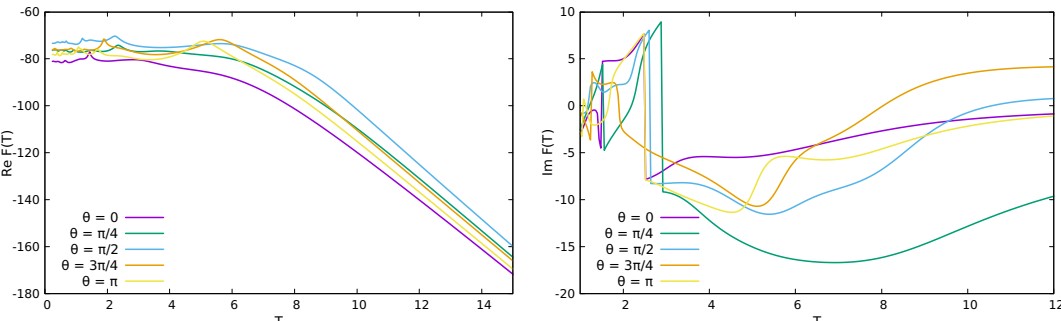

Figure 13: Free energy for $N = 34$, $r = 2$ and different $\theta$'s. The random couplings are the same for all angles. Qualitatively, results do not depend much on $\theta$, especially those corresponding to the real part of the free energy.

gravity results where we also observe a maximum $j_0$ for half-wormholes to exist. As mentioned earlier, we do not try to make any quantitative relation between the SYK parameter $r$ and the gravitational parameter $j_0$ but we compare the qualitative behavior.

The imaginary part of the free energy also reveals interesting features. For small $r$, see Fig. 11, there is an intricate oscillating pattern that seems to be very sensitive to the details of the random couplings. For sufficiently high temperature, the imaginary part vanishes as it was the case in the gravity calculation. For larger $r$, a saw-like structure robustly emerges that does not depend on the details of the random couplings. For intermediate values of $r$, the saw-like shape of the imaginary part coexists with a flat part in the low temperature limit of the real free energy. This is in striking agreement with the gravity calculation described in section 4. Moreover, the saw-like shape is well described by the gravitational prediction (51). see Fig. 12. The value of the fitting parameters could be employed to establish an effective relation between $j(\tau)$ in gravity and $z$ in the SYK model.

It seems that this range of intermediate $z$ is the one closely related to the gravitational half-wormhole. For larger $r$, the saw-like shape covers a broad range of temperatures while the real part no longer has a half-wormhole contribution. This is for a fixed $\theta = \pi/6$. Results depicted in Fig. 13 indicate that for a fixed $r$, and varying $\theta$, the free energy, especially the real part, is qualitatively similar. For small $\theta \ll \pi/2$ (not shown), the free energy does not change much with $\theta$. As $\theta$ increases, the details of the oscillations of the imaginary part for low temperatures become very sensitive to the value of $\theta$. However, the overall pattern does not change substantially. More specifically, the saw-like shape that defines the large $r$ limit of the imaginary free energy cannot be reached by tuning $\theta$. Likewise, the overall shape of the real part of the free energy is rather insensitive to $\theta$ though the position of the local fluctuation does change with it. We note that the same random couplings were employed for all values of $\theta$.

Previously, we found that for $z = 0$ results do not depend qualitatively on the disorder realization. We now repeat this analysis for $z \neq 0$. Since results do not depend qualitatively on $\theta$, we set $\theta = \pi/6$ and study the free energy for two different disorder realizations $a, b$ and different $r$'s. For small $r$, results are not very different from the $r = 0$ case. The real free energy is rather similar for both disorder realizations with a different pattern of small fluctuations in the low temperature flat piece. The oscillating pattern of the imaginary part for small temperature seems to be more sensitive to the choice of random couplings. However, as $r$ increases, both the real and imaginary part become increasing insensitive to the disorder realization. This is hardly surprising since the value of $z$ becomes much larger than the typical value of the random coupling so disorder becomes less important. The vanishing of the wormhole phase for large $r$, even in the low temperature limit, is consistent with previous results

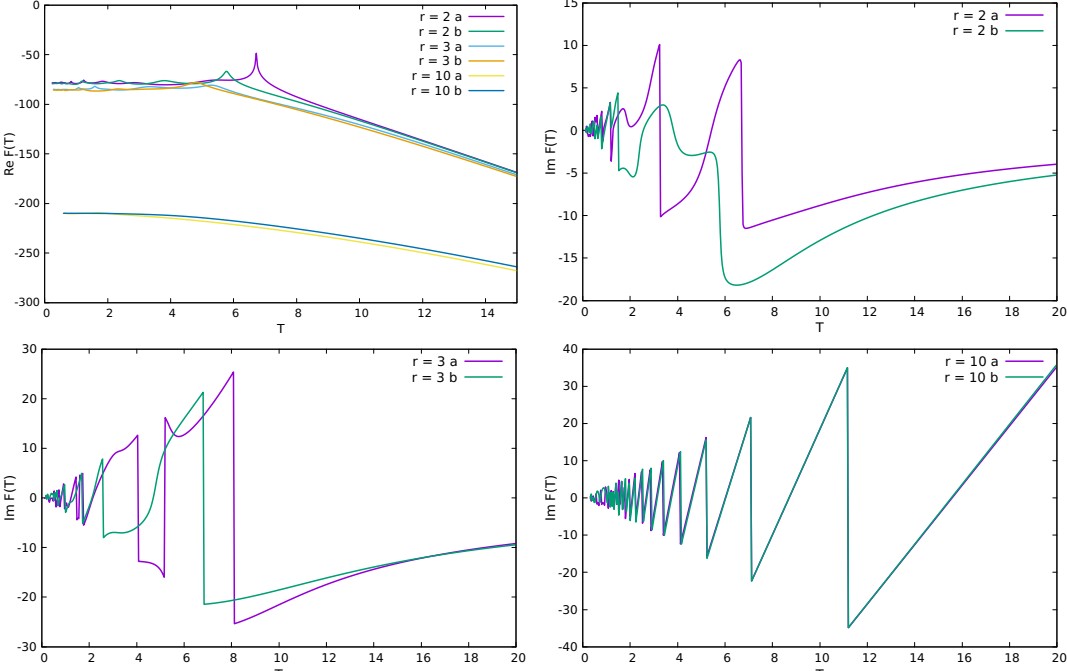

Figure 14: Free energy for $N = 34$, two different disorder realizations $(a, b)$, $\theta = \pi/6$ and different $r$'s. Top left is the real part. The rest correspond to the imaginary part. In all cases, results for the two different disorder realizations are qualitatively similar. As $r$ increases, the dependence on the specific choice of random couplings become smaller.

as well.

This is just an exploratory study and presently we do not have a very precise understanding of the reasons behind the surprising agreement with the half-wormhole gravity prediction in the intermediate range of parameters mentioned above. It would be interesting to carry out a more precise analysis along the lines of [35].

In summary, we have shown that the half-wormhole appears to capture a single realization of the SYK model with complex couplings. In some range of parameters, we have found a surprising agreement in the free energy between both models. This is still a rather phenomenological approach. It would be interesting to check whether the effective Schwarzian-like action is the same in both models or whether other characterizations of the single realization beyond $z$ give similar agreement.

Finally, let us comment on performing the SYK average and its relation to the gravity average discussed in section 5. In the SYK ensemble, $J_{ijk\ell}$ and $M_{ijk\ell}$ are Gaussian random variable with zero mean and standard deviation $(12/N)^{3/2}$. This ensemble will give some distribution for the parameter $z$. As we expect the parameter $z$ to correspond to $j_0$ in gravity, we see that the dual picture corresponds to performing the average over $j_0$ where its standard deviation $J$ is fixed to some value, related to the standard deviation of $z$ in the SYK ensemble. This is different from the simple gravity regime which corresponds to $J \to +\infty$. In fact, as $z$ is very small in a typical realization, we expect that $J$ should be very small. The quenched average will preserve the real part and cancel out the imaginary part as shown in section 5.2.3. Thus, the basic features of the quenched free energy will not depend very much on the value of $J$ and the agreement described in section 2 will be robust, and holds at $J \to +\infty$ but also at small $J$.

# 7 Discussion

We have given a precise interpretation of a Euclidean wormhole in a "simple" gravity theory as an average of half-wormholes in "exact", or non-averaged, theories. The exact theory, which manifestly factorizes, is supposed to capture some of the complexity of the UV-completion of gravity. This interpretation shows that the wormhole is meaningful and captures statistical properties of objects in some exact theory. We will now mention a few interesting implications of our results.

**Dissecting the average.**   Our result suggests a systematic procedure to improve simple gravity towards a more "exact" theory of gravity. Given some simple theory of gravity, one would imagine having many types of wormholes each leading to a factorization puzzle. Replacing any of these wormholes by half-wormholes would bring us closer and closer to the exact theory in which factorization is manifest. It would be interesting to understand if this procedure can be implemented for other theories of interest.

**Relation to baby universes.**   Another approach to the factorization problem is to use baby universes and $\alpha$-states [7,9,49–52,61]. Our discussion here will be placed in the framework of [50]. In the regime of interest, the simple gravity theory has only the black hole and wormhole saddle-points. From this, it can be shown that the third-quantized theory is Gaussian and that the baby universe Hilbert space is a Fock space. Explicit expressions for the $\alpha$-states can be obtained. An $\alpha$-state is implemented by adding an infinite number of SD-branes and factorization in an $\alpha$-state can be understood as the consequence of a "wormhole=diagonal" identity [49,51,61]. See Appendix B for more details.

   Despite some similarities, we view our resolution of the factorization problem as conceptually different. We are not adding anything to simple gravity to restore factorization but are rather interpreting the wormhole as the average of half-wormholes belonging to non-averaged theories.[7] Our approach is more economical because factorization is achieved with a single SD-brane whereas an $\alpha$-state corresponds to an infinite number of them. Moreover, our SD-branes are qualitatively different from the SD-branes of [50] since they don't involve the asymptotic boundary but attach to the geometry deep inside the bulk. Also, there is a priori no conceptual difficulty in constructing higher-dimensional versions of our half-wormholes whereas the relevance of baby universes in higher dimensions is subject of debate as, among other things, they don't fit well with string theory [62].

**Higher dimensions.**   Let us now comment on possible generalizations to higher dimensions. At the difference of [16], where the full path integral was performed, we are using here a sum over saddle-points.[8] In higher dimensions, we also don't expect the full path integral to be meaningful and should be defined as a sum over saddle-points. This makes the JT model with matter more realistic than pure JT gravity.

   Our half-wormhole solutions seem to be generalizable to higher dimensions. For example, one could try to construct half-wormholes using the the Euclidean wormholes solutions described in [31] which are very similar to our Euclidean wormhole. Indeed, they are also supported by boundary sources and display a similar phase transition at low temperature. It would be interesting to see if consistent boundary conditions can be imposed at the mid-section to obtain higher-dimensional half-wormholes.

---

[7]That said, it has been pointed out in [35] that multiple bulk descriptions could coexist.

[8]This is necessary because the full path integral is divergent. For example, the integral over $b$ in the sum over double trumpet geometry diverges due to the Casimir energy of the matter.

Assuming that Euclidean wormholes which are asymptotically $\text{AdS}_5 \times S^5$ do make sense, they will lead to a factorization problem. Our proposal suggests that it should be possible to define half-wormholes with boundary condition $j(\tau)$ so that the wormhole arises meaningfully in the theory where we average over $j(\tau)$. This would provide a bulk resolution of the factorization problem.

As the SYK model is defined as an average, it is natural to take the exact gravity theory to be dual to a single realization of the couplings, as we have done in this paper. In the case of $\text{AdS}_5 \times S^5$, it's much less clear what we should do with the dual $\mathcal{N} = 4$ super Yang-Mills. Our proposal suggests that there should be some specific modification of $\mathcal{N} = 4$ SYM which should be dual to having an SD-brane with boundary condition $j(\tau)$ in the bulk. Averaging over such modifications will give a theory whose gravity dual has the Euclidean wormhole. It would be interesting to see if this idea can be made more precise.

## Acknowledgments

AMG acknowledges support from a personal NSFC Grant No. 11874259 (AMG), the National Key R&D Program of China (Project ID: 2019YFA0308603) and a Shanghai talent program. VG acknowledges the postdoctoral program at ICTS for funding support through the Department of Atomic Energy, Government of India, under project no. RTI4001.

## A  Wormhole dynamics

The Euclidean wormhole has two reparametrization modes $\tau_L(u_L)$ and $\tau_R(u_R)$ on each side of the double trumpet. In this section, we derive the Schwarzian-like effective action for these modes. This analysis shows that the wormhole is entirely a consequence of the U(1) gauge constraint. It also allows us to derive the spectrum of excitations of the wormhole.

The effective action for $\tau_L(u_L)$ and $\tau_R(u_R)$ takes the form

$$S_{\text{JT}} = -\phi_r \int du_L \left\{ \tanh\left(\tfrac{1}{2}\tau_L(u_L)\right), u_L \right\} - \phi_r \int du_R \left\{ \tanh\left(\tfrac{1}{2}\tau_R(u_R)\right), u_R \right\} + S_\chi, \tag{95}$$

where the two Schwarzian terms come from the JT action. The interesting piece is the contribution $S_\chi$ of the scalar field. It is obtained by writing the profile of the scalar field in terms of boundary sources $\chi_L$ and $\chi_R$:

$$\chi(\tau, \rho) = \int_{-\infty}^{+\infty} d\tau_L \, K_L(\tau, \rho; \tau_L) \chi_L(\tau_L) + \int_{-\infty}^{+\infty} d\tau_R \, K_R(\tau, \rho; \tau_R) \chi_R(\tau_R), \tag{96}$$

where the bulk-to-boundary propagators on global $\text{AdS}_2$ are

$$K_L(\tau, \rho; \tau_L) = \frac{1}{2\pi} \left( \frac{\cos\rho}{\cosh(\tau - \tau_L) + \sin\rho} \right), \quad K_R(\tau, \rho; \tau_R) = \frac{1}{2\pi} \left( \frac{\cos\rho}{\cosh(\tau - \tau_R) - \sin\rho} \right).$$

While the sources $\chi_L$ and $\chi_R$ are periodic under $\tau \sim \tau + b$, the integral in (96) is over the real line to implement the method of images. This leads to

$$S_\chi = -\frac{1}{4\pi} \int_0^{\phi_r \beta} du_L \int_{-\infty}^{+\infty} du_R \frac{\tau_L'(u_L) \tau_R'(u_R)}{\cosh^2\left(\tfrac{1}{2}(\tau_L(u_L) - \tau_R(u_R))\right)} \chi_L(u_L) \chi_R(u_R), \tag{97}$$

where we ignore the left-left and right-right terms that are not important for our analysis.

The choice of sources corresponding to our wormhole is

$$\chi_L(u_L) = -ik, \qquad \chi_R(u_R) = ik. \tag{98}$$

For this choice, it can be seen that the integrand of $S_\chi$ becomes a total derivative. This is expected because there are no interaction between the two sides. Instead, we will see that the dynamics comes entirely from a gauge constraint.

In JT gravity, we have to view the $SL(2,\mathbb{R})$ symmetry as a gauge symmetry because we should not sum over equivalent configurations in the path integral [33]. The vanishing of $SL(2,\mathbb{R})$ charge was used in [53] to study the eternal traversable wormhole and to reduce to the dynamics of a Liouville particle. In the double trumpet geometry, the $SL(2,\mathbb{R})$ symmetry of global $AdS_2$ is broken to $U(1)$ because of the identification $\tau \sim \tau + b$.

The $U(1)$ symmetry acts on the reparametrization modes as

$$\tau_L(u_L) \to \tau_L(u_L) + \varepsilon, \qquad \tau_R(u_R) \to \tau_R(u_R) + \varepsilon, \tag{99}$$

where $\varepsilon$ is a constant. To obtain the $U(1)$ charges, we make $\varepsilon$ dependent on $u_L$ and $u_R$. The charges $Q_L$ and $Q_R$ can be read off from the variation of the action

$$\delta_\varepsilon S = \int du_L du_R \left( \partial_{u_L} \varepsilon(u_L, u_R) Q_L(u_L) + \partial_{u_R} \varepsilon(u_L, u_R) Q_R(u_R) \right). \tag{100}$$

We have a $U(1)$ charge at each boundary taking the form

$$\begin{aligned}
Q_L(u_L) &= -\phi_r \left( \frac{\tau_L^{(3)}(u_L)}{\tau_L'(u_L)^2} - \frac{\tau_L''(u_L)^2}{\tau_L'(u_L)^3} - \tau_L'(u_L) \right) - \frac{k^2}{\pi}, \\
Q_R(u_R) &= -\phi_r \left( \frac{\tau_R^{(3)}(u_R)}{\tau_R'(u_R)^2} - \frac{\tau_R''(u_R)^2}{\tau_R'(u_R)^3} - \tau_R'(u_R) \right) - \frac{k^2}{\pi}.
\end{aligned} \tag{101}$$

and a similar expression at the right boundary. The gauge constraint condition is then

$$Q_L(u_L) + Q_R(u_R) = 0. \tag{102}$$

As these are functions of different variables, the most general solution is

$$Q_L = -q, \qquad Q_R = q, \tag{103}$$

where $q$ is some constant. We observe that this is equivalent to two decoupled particles with correlated potentials.

Indeed, using Liouville variables $\varphi_{L/R}(u) = \log \tau_{L/R}'(u)$, the action takes the form

$$S_{\text{Liouville}} = \phi_r \int du \left[ \frac{1}{2} \varphi_L'^2(u) + \frac{1}{2} \varphi_R'^2(u) + V_q(\varphi_L) + V_{-q}(\varphi_R) \right], \tag{104}$$

with the Liouville potential

$$V_q(\varphi) = \frac{1}{2} e^{2\varphi} - \frac{1}{\phi_r} \left( \frac{k^2}{\pi} - q \right) e^\varphi. \tag{105}$$

This is rather similar to the Liouville particle description of the eternal traversable wormhole [53]. The minimum of the potential corresponds to the solutions

$$\tau_L(u) = \frac{1}{\phi_r} \left( \frac{k^2}{\pi} - q \right) u, \qquad \tau_R(u) = \frac{1}{\phi_r} \left( \frac{k^2}{\pi} + q \right) u, \tag{106}$$

which are precisely the solutions corresponding to the wormhole [48]. We also see that we have

$$q = \frac{\eta}{2}, \tag{107}$$

in terms of the asymmetry parameter $\eta$.

In the range $|q| < k^2/\pi$, we have bound states corresponding to the low energy excitations of the wormhole. Small fluctuations around the minimum gives two oscillators with frequencies

$$\omega_L = \frac{1}{\phi_r}\left(\frac{k^2}{\pi^2} - q\right), \qquad \omega_R = \frac{1}{\phi_r}\left(\frac{k^2}{\pi^2} + q\right). \tag{108}$$

In particular, the energy gap is

$$E_{\text{gap}} = \frac{1}{2}(\omega_L + \omega_R) = \frac{2k^2}{\phi_r \pi^2}. \tag{109}$$

The $k$-dependence of the energy gap matches the corresponding quantity in the dual SYK model, as was shown in [58] by an analysis of the Schwinger-Dyson equation.

## B  Baby universes and $\alpha$-states

The gravity model considered in this paper turns out to be a nice example where the framework of [50] can be implemented explicitly. We define here the gravitational path integral to be the sum over saddle-points. Here, we are only discussing the "simple" gravity theory with the black hole and wormhole saddle-points.

At each boundary, we can choose the inverse temperature and the boundary source $k$ for the scalar field: $\chi|_\partial = ik$. For suitable regimes of these parameters, there are only two saddle-points: the black hole and the wormhole.

The connected two-point function is the wormhole partition function

$$\langle Z(J_1)Z(J_2)\rangle_c = Z_{\text{WH}}(J_1, J_2), \tag{110}$$

where at each boundary we have the label $J = (\beta, k)$. All the connected higher-point functions vanish. Indeed, although multiboundary saddle-points could exist in principle, we can safely neglect them because they are exponentially suppressed by $S_0$. Although it won't be needed in this discussion, let us report that the explicit expression is

$$Z_{\text{WH}}(J_1, J_2) = z_{\text{WH}} \exp\left(\frac{(k_1 - k_2)^4}{8\pi(T_1 + T_2)}\right), \tag{111}$$

where the prefactor is given in (6) with $b = \frac{1}{\pi T}(k_1 - k_2)^2$ and $T = \frac{1}{2}(T_1 + T_2)$.

The fact that only the one-point and two-point function are non-zero tells us that $Z(J)$ is a Gaussian random variable. This theory fits in the "wormhole perturbation theory" discussed in [50]. In fact, the third-quantized theory is free here.

To be more explicit, we should make a change of variable and consider

$$\widetilde{Z}(J) = \int dJ' \, G(J, J')(Z(J') - \langle Z(J')\rangle), \tag{112}$$

where $G(J, J')$ is a suitable kernel that diagonalizes $Z_{\text{WH}}$ according to

$$\int dJ_1' dJ_2' \, G(J_1, J_1')G(J_2, J_2')Z_{\text{WH}}(J_1', J_2') = \delta(J_1 - J_2). \tag{113}$$

The existence of $G$ can be seen as follows. We can view $Z_{\text{WH}}(J_1, J_2)$ as an operator on the space of functions $\{f(J)\}$ defined by $(Z_{\text{WH}} \cdot f)(J) = \int dJ' Z_{\text{WH}}(J, J') f(J')$. As this operator is symmetric, it can be diagonalized. This means that we can find another operator $G$ such that $G^t \cdot Z_{\text{WH}} \cdot G = \mathbf{1}$. We have also removed the one-point function in (112) for convenience.

We then see that we have

$$\langle \widetilde{Z}(J_1) \widetilde{Z}(J_2) \rangle = \delta(J_1 - J_2). \tag{114}$$

Following [50], we can write

$$\widetilde{Z}(J) = a_J + a_J^\dagger, \tag{115}$$

where $a_J, a_J^\dagger$ are baby universe annihilation and creation operators satisfying

$$[a_{J_1}, a_{J_2}^\dagger] = \delta(J_1 - J_2). \tag{116}$$

The baby universe Hilbert space is then the Fock space generated by acting with $a_J^\dagger$ on the Hartle-Hawking vacuum $|\text{HH}\rangle$, see [50] for more details. An $\alpha$-state $|\alpha\rangle$ is specified by an arbitrary function $\alpha(J)$ and satisfies

$$\widetilde{Z}(J)|\alpha\rangle = \alpha(J)|\alpha\rangle. \tag{117}$$

They can be written explicitly as coherent states of baby universes

$$|\alpha\rangle = \mathcal{N} \exp\left[ -\frac{1}{2} \int d\tau (a_J^\dagger - \alpha(J))^2 \right]. \tag{118}$$

We can also write them as the Fourier transforms of SD-brane states according to [50]

$$|\alpha\rangle = \int Dg(J) \exp\left( -i \int dJ\, g(J)\alpha(J) \right) |\text{SD}_g\rangle, \tag{119}$$

where $|\text{SD}_g\rangle = \exp\left( i \int dJ\, g(J) Z(J) \right) |\text{HH}\rangle$ has the interpretation of the insertion of an SD-brane. This works because the integral over $g(J)$ gives a delta functional $\delta(Z(J) - \alpha(J))$ projecting onto the $\alpha$-state $\alpha(J)$. Note that the SD-branes discussed here have nothing to do with the SD-brane used to define the half-wormhole: they involve the asymptotic boundary whereas our SD-brane attaches to the small end of the trumpet.

An $\alpha$-state is implemented in gravity by adding an infinite number of SD-branes. Factorization in an $\alpha$-state can then be understood in terms of a "wormhole=diagonal" identity [49,51]. In the present case, this identity can be written explicitly. This analysis is identical to the corresponding analysis in the $\widehat{\text{CGHS}}$ model [61,63,64], a flat space analog of JT gravity in which $Z(\beta)$ is a Gaussian variable. We refer to [61] for more details.

# C  Some technical details

## C.1  Variational problem

We check here that the boundary conditions described in (3.1) are consistent with the variational problem. The JT action is

$$I_{\text{JT}} = -\frac{1}{2} \int d^2x \sqrt{g}\, \Phi(R + 2) + I_\partial, \tag{120}$$

where $I_\partial$ is a suitable boundary term to be determined. The variation gives

$$\delta I_{\text{JT}} = -\frac{1}{2} \int d^2x \sqrt{g} \Big( (R+2)\delta\Phi + \frac{1}{2} g^{\mu\nu}\Phi(R+2)\delta g_{\mu\nu} + \Phi(-R^{\mu\nu} + \nabla^\mu\nabla^\nu - g^{\mu\nu}\Box)\delta g_{\mu\nu} \Big) + \delta I_\partial \,. \tag{121}$$

Imposing the dilaton equation of motion gives $R = -2$ and we get

$$\delta I_{\text{JT}} = -\frac{1}{2} \int d^2x \sqrt{g} \Big( \Phi(g^{\mu\nu} + \nabla^\mu\nabla^\nu - g^{\mu\nu}\Box)\delta g_{\mu\nu} \Big) + \delta I_\partial \,. \tag{122}$$

Performing integration by parts twice yields

$$\delta I_{\text{JT}} = \int d^2x \Big( E_g^{\mu\nu}\delta g_{\mu\nu} + \partial_\mu\Theta^\mu \Big) + \delta I_\partial \,, \tag{123}$$

where

$$\Theta_\mu = -\frac{1}{2}\sqrt{g} \Big( \Phi\nabla^\nu\delta g_{\mu\nu} - \Phi g^{\alpha\beta}\nabla_\mu\delta g_{\alpha\beta} - \nabla^\nu\Phi\delta g_{\mu\nu} + g^{\alpha\beta}\nabla_\mu\Phi\delta g_{\alpha\beta} \Big) \,. \tag{124}$$

The geodesic boundary at $\rho = 0$ will be denoted $\gamma$. The normal vector is $n = \partial_\rho$ and the extrinsic curvature vanishes there. Our choice of boundary condition forces that $\delta K = 0$ at $\gamma$. As we have

$$\delta K = -n^\mu\nabla^\nu\delta g_{\mu\nu} + \frac{1}{2} n^\alpha\nabla_\alpha\delta g_{\mu\nu}g^{\mu\nu} \,, \tag{125}$$

we obtain

$$n^\mu\Theta_\mu|_\gamma = \frac{1}{2}\sqrt{g}\, n^\mu \Big( \nabla^\nu(\Phi\delta g_{\mu\nu}) - g^{\alpha\beta}\nabla_\mu\Phi\delta g_{\alpha\beta} \Big) \,. \tag{126}$$

The first term becomes a total derivative on the circle. Thus, we end up with

$$\int_\gamma n^\mu\Theta_\mu = -\frac{1}{2}\int d\tau\, \partial_\rho\Phi\, g^{\alpha\beta}\delta g_{\alpha\beta} \,. \tag{127}$$

To have a consistent variation problem, this should be cancelled by the boundary term

$$I_\partial = \int_\gamma d\tau \sqrt{g}\, n^\mu\partial_\mu\Phi = \int_\gamma d\tau\, \partial_\rho\Phi \,, \tag{128}$$

using that $\delta(\sqrt{g}) = \frac{1}{2}\sqrt{g}\, g^{\alpha\beta}\delta g_{\alpha\beta}$. Note that for the end-of-the-world brane with condition $n^\mu\partial_\mu\Phi = \mu$, this reproduces the known action for an end-of-the-world brane, used for example in [18]:

$$I_\partial = \mu\int_\gamma d\tau \,. \tag{129}$$

For the scalar field, we must impose

$$n^\mu\partial_\mu\chi\,\delta\chi|_\gamma = 0 \,, \tag{130}$$

so its consistent to take our Dirichlet boundary conditions which correspond $\delta\chi = 0$.

For consistency with the JT equations of motion, we cannot impose $n^\mu\partial_\mu\Phi = \mu$ for constant $\mu$. Indeed, one of the JT equations says that

$$\partial_\tau\partial_\rho\Phi + \partial_\rho\chi\partial_\tau\chi = 0 \,. \tag{131}$$

So if we want $\chi$ to be non-trivial on $\gamma$, we should not impose any local condition on the dilaton.

It's not difficult to solve the JT equations for a general $j(\tau)$ in an expansion around $\rho = 0$ to see that our boundary conditions are consistent. We also note that the additional boundary action $I_\partial$ is just the zero mode of $\partial_\rho \Phi$. This zero mode is not fixed by the equations of motion. In the wormhole, it corresponds to the asymmetry parameter $\eta$ discussed in [48] which does not play an important role. For simplicity, we will set this zero mode to zero as an additional boundary condition. This ensures that

$$I_\partial = 0 \tag{132}$$

and we don't have to add an additional boundary term to the action.

### C.2   Scalar profile in half-wormhole

We would like to compute the on-shell action of the scalar field in the half-wormhole. The boundary conditions are

$$\chi|_{\rho=0} = j(\tau), \qquad \lim_{\rho \to \frac{\pi}{2}} \chi = ik. \tag{133}$$

To do this, we can use the general expression (96) of the scalar field in the double trumpet. This leads to

$$\chi(\tau, \rho) = \frac{2ik}{\pi} \rho + \int_{\mathbb{R}} d\tau_L K_L(\tau, \rho; \tau_L) \chi_L(\tau_L), \tag{134}$$

where the method of images has been implemented by extending the range of integration of $\tau_L$ to the full real line.

Of course, we don't really have a left source $\chi_L$ here, it's just a convenient way to write a solution of the Laplace equation. We can find the relation between $\chi_L(\tau)$ and $j(\tau)$ by going to Fourier space. We obtain

$$\chi_L^{(n)} = 2 \cosh(\tfrac{\pi^2 n}{b}) j_n. \tag{135}$$

The on-shell action is given by

$$S^\chi_{\text{on-shell}} = \frac{1}{2} \int_M d^2x \sqrt{g} (\partial \chi)^2 = \frac{1}{2} \int_{\partial M} dy \sqrt{h} \, \chi \, n^\mu \partial_\mu \chi = \frac{1}{2} \int_0^b d\tau \left[ \chi \partial_\rho \chi \right]_{\rho=0}^{\rho=\frac{\pi}{2}}, \tag{136}$$

which can be written as

$$S^\chi_{\text{on-shell}} = -\frac{k^2 b}{\pi} + S_1 + S_2, \tag{137}$$

in terms of a linear in $j(\tau)$ piece $S_1$ and a quadratic piece $S_2$. The linear piece is

$$\begin{aligned} S_1[j(\tau)] &= -\frac{ik}{\pi} \int_0^b d\tau \int_{\mathbb{R}} d\tau_L \chi_L(\tau_L) \left( \frac{1}{4(1 + \cosh(\tau - \tau_L))} + \frac{1}{2\pi \cosh(\tau - \tau_L)} \right) \\ &= -\frac{2ikb}{\pi} j_0, \end{aligned} \tag{138}$$

where we have used (135). The quadratic piece is

$$\begin{aligned} S_2[j(\tau)] &= \frac{1}{8\pi^2} \int_0^b d\tau \int_{\mathbb{R}} d\tau_L \int_{\mathbb{R}} d\tau_L' \frac{\chi_L(\tau_L) \chi_L(\tau_L')}{\cosh^2(\tau - \tau_L) \cosh(\tau - \tau_L')} \\ &= \frac{1}{4\pi} \int_0^b d\tau \int_{\mathbb{R}} d\tau_L \frac{\chi_L(\tau_L) j(\tau)}{\cosh^2(\tau - \tau_L)}. \end{aligned} \tag{139}$$

In Fourier space, this gives

$$S_2[j(\tau)] \;=\; \frac{1}{2\pi} \sum_{n \in \mathbb{Z}} \frac{\pi^2 n}{\sinh(\frac{\pi^2 n}{b})} \chi_{L,n} j_{-n}\,, \tag{140}$$

where we used that

$$\int_{\mathbb{R}} dx\, \frac{e^{i\omega x}}{\cosh^2 x} = \frac{\pi \omega}{\sinh(\frac{1}{2}\pi\omega)}\,. \tag{141}$$

Finally, using the relation (135), we obtain the Fourier representation

$$\begin{aligned}
S_2[j(\tau)] &= \sum_{n \in \mathbb{Z}} \frac{\pi n}{\tanh(\frac{\pi^2 n}{b})} |j_n|^2 \\
&= \frac{b}{\pi} j_0^2 + 2 \sum_{n \geq 1} \frac{\pi n}{\tanh(\frac{\pi^2 n}{b})} |j_n|^2\,.
\end{aligned} \tag{142}$$

We finally arrive at

$$S^\chi \;=\; -\frac{b}{\pi}(k + i j_0)^2 + 2 \sum_{n \geq 1} \frac{\pi n}{\tanh(\frac{\pi^2 n}{b})} |j_n|^2\,. \tag{143}$$

## C.3   Product identity for the average

We will derive here the identity (67). Let's introduce the variable

$$q = e^{2i\pi\tau} = e^{-4\pi^2/b}\,. \tag{144}$$

We use the standard notation $\tau$ for an element of the upper half-plane $\operatorname{Im} \tau > 0$. This should not be confused with the Euclidean time coordinate which plays no role in this section. This allows us to write

$$\tanh\left(\frac{\pi^2 n}{b}\right) = \frac{q^{-n/4} - q^{n/4}}{q^{-n/4} + q^{n/4}}\,, \tag{145}$$

and we have

$$\prod_{n \geq 1} \tanh\left(\frac{\pi^2 n}{b}\right) = \prod_{n \geq 1} \frac{1 - q^{n/2}}{1 + q^{n/2}} = \prod_{n \geq 1} \frac{(1 - q^{n/2})^2}{1 - q^n} = \frac{\eta(\tau/2)^2}{\eta(\tau)}\,, \tag{146}$$

in terms of the Dedekind eta function $\eta(\tau)$. The large $b$ regime corresponds to $q \to 1$. In this regime, it is natural to apply a modular S-transform and define

$$\tilde{\tau} = -\frac{1}{\tau}\,, \qquad \tilde{q} = e^{2i\pi\tilde{\tau}} = e^{-b}\,, \tag{147}$$

so that we can use the transformation law $\eta(-1/\tau) = \sqrt{-i\tau}\, \eta(\tau)$ to obtain

$$\prod_{n \geq 1} \tanh\left(\frac{\pi^2 n}{b}\right) = 2\sqrt{-i\tilde{\tau}}\, \frac{\eta(2\tilde{\tau})^2}{\eta(\tilde{\tau})} = 2\tilde{q}^{1/8} \sqrt{-i\tilde{\tau}} \prod_{n \geq 1} \frac{(1 - \tilde{q}^{2n})^2}{1 - \tilde{q}^n}\,, \tag{148}$$

which is (67).

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
