# Peer review of "Half-wormholes in nearly AdS2 holography"

_SciPost Physics, doi:SciPost Phys. 12, 135 (2022)_

## Round 1 · Referee Report · Anonymous (Referee 1) · 2021-10-25

Report

The paper studied half-wormhole configurations in a varian of SYK/JT with complex couplings. Despite the couplings being complex, the authors argued in their previous paper that the theory still retains certain hermitian properties and the partition function is well defined. The major interesting point of this model is that wormhole configurations can be easily seen. In the current paper they continue their investigation in order to find half-wormhole configurations.

The authors propose to introduce a brane in the AdS_2 which is dual to a particular choice of couplings on the SYK side. They proceed by studying the corresponding half-wormhole configurations in detail and demonstrate how they affect the factorization. Also they present some numerical calculations in SYK to support their proposal.

The paper is clearly written and is definitely interesting for the field, so I recommend it for publication. However, I believe a few questions should be clarified: My main confusion is related to the quantity $z$ introduced in eq. (6.2). As far as I understand, SYK hamiltonian (6.1) depends only on the antisymmetric part of $J_{ijkl}, M_{ijkl}$. $z$ can only introduce an overall shift in the energy.

Also I have a few minor questions: 1. In Secton 2.1 it would be useful to emphasize whether there is a single complex field $\chi$ or a large number N of them. I expect that there should be a large number in order to compete with gravity fluctuations and produce a large determinant in Section 3.2.2 2. As far as I can understand, the nice equation (5.16) does not rely on a particular choice of $j(\tau)$ action( (5.3) or (5.4) ). I think it would be very helpful for readers if it is emphasized around eq. (5.16). 3. Finally, it would be very interesting if the authors could comment whether something like (5.16) would hold for correlation functions. From conservative point of view, one could argue that the measure for $j(\tau)$ can always be fine-tuned to get (5.16), but matching the correlation function is non-trivial.

  • validity: -
  • significance: -
  • originality: -
  • clarity: -
  • formatting: -
  • grammar: -

Author:  Victor Godet  on 2022-01-15  [id 2097]

(in reply to Report 1 on 2021-10-25)

We would like to thank the referee for his comments.

The quantity $z$ should be viewed as a coarse-grained version of a single realization $J_{ijkl}$ of the SYK couplings. We claim that the half-wormhole should be dual to a single realization of the SYK couplings, the choice of $J_{ijkl}$ corresponding to a choice of boundary condition $j(\tau)$. In trying to make this more quantitative, it is certainly be too ambitious to try to reproduce the exact value of $J_{ijkl}$ in gravity. For this reason, we consider a coarse-grained version: the "center-of-mass" $z$ obtained by summing over $i,j,k,l$. As we argued in our paper, $z$ is dual to the zero mode $j_0$ of $j(\tau)$ and we give concrete numerical evidence for this. This is our motivation to introduce the quantity $z$.

  1. A single scalar field is enough in our case. This is because our wormhole is a classical solution, supported by a classical profile for the scalar field. This is different than the Maldacena-Qi wormhole where the wormhole is supported by quantum effects so that a large number of scalar fields is required. In other words, the scalar contribution is not small but proportional to $k$, which we can take as large as we want.
  2. This is correct. The fact that the wormhole emerges from the average of two half-wormholes is a general fact of the "uniform limit" $J\to+\infty$ which doesn't depend on details of the ensemble for $j(\tau)$. We have added a sentence after (5.17) to emphasise this. Note that the following paragraph explains why this is the case from a path integral point of view.
  3. The fact that the wormhole emerges from the average of half-wormholes (equation (5.16)) would also work for correlation functions. As explained below Figure 6, our average has the simple interpretation of "finishing the path integral". That is, the path integral on the wormhole can be done in two steps: we can first integrate over all fields except the value $j(\tau)$ of the scalar at the geodesic boundary. Interpreting the result as the contributions of two half-wormholes, the final integral over $j(\tau)$ is then interpreted as an average over half-wormhole boundary contributions. This interpretation shows that this will also work for more general observables.

---

## Round 1 · Referee Report · Anonymous (Referee 2) · 2021-11-22

Report

This paper studies the wormhole and half-wormhole solutions in JT gravity coupled to a complex scalar field, and also proposes a possible interpretation of these solutions in the SYK model. The topic of the paper is interesting and the results are new and well written. Therefore, I recommend for publication in SciPost Physics once the authors address the following points. \\

$\bullet$ The complex couplings introduced in (2.13), (2.14) seem to give non-unitary evolution for each SYK system, even though the total Hamiltonian (2.12) is Hermitian. It might be explained in the earlier paper, but it's useful to explain again how the authors justify these type of couplings in that section. \\

$\bullet$ The SYK model has the zero mode sector which is described by the Schwarzian theory and dual to the pure JT gravity. The SYK model also has the non-zero mode sector, which seems to give an infinite number of matter fields in the dual gravity theory. In this paper, the authors consider a single scalar field (2.2) in the gravity theory. It's useful to explain how the authors think of this single scalar field (2.2) from the SYK point of view. (i.e. do they correspond to a particular non-zero mode in the SYK or any kinds of external degree of freedom?)\\

$\bullet$ Even though the pure JT gravity (without matter) is one-loop exact, I don't think the system of JT gravity coupled to a matter is one-loop exact. The interactions between the Schwarzian mode and the matter field (like (A.3)) give contributions for two-loop and higher.\\
  • validity: -
  • significance: -
  • originality: -
  • clarity: -
  • formatting: -
  • grammar: -

Author:  Victor Godet  on 2022-01-15  [id 2098]

(in reply to Report 2 on 2021-11-22)

We would like to thank the referee for his comments.

  • We have added a paragraph below 2.11 to explain the physical significance of the imaginary sources as explained in our earlier paper.
  • We view the massless scalar field as modeling the complicated SYK deformation corresponding to adding an imaginary part to the coupling. As the referee points out, this deformation should really turn on sources for a tower of bulk fields with increasing masses. Interestingly, a single massless scalar field seems to be enough to capture the wormhole physics in our regime. One way to think about this is that although many fields will be turned on in the true dual of SYK, most of these are irrelevant and the main effect is captured by a marginal operator dual to a single massless scalar field $\chi$. Given that the dual of SYK is not well understood, we have not tried to make this intuition more precise. We have added a paragraph around equation (2.17) to explain this.
  • We only consider the theory with constant boundary sources in which case it is one-loop exact (in particular (A.3) becomes a total derivative). This is just to emphasize that all the partition functions we compute are one-loop exact. We have added "with constant boundary sources" at the beginning of section 3.2.2. to clarify this.

Anonymous on 2022-02-02  [id 2146]

(in reply to Victor Godet on 2022-01-15 [id 2098])

Satisfactory addressed

---

## Round 2 · Referee Report · Anonymous (Referee 1) · 2022-1-22

Report

What I wanted to say is that $z$ is an overall shift in the SYK definition of energy: for a generic tensor $J_{ijkl}$, the sum $\sum_{ijkl} J_{ijkl} \psi_i \psi_k \psi_k \psi_l$ can be separated into two parts. The antisymmetric part of $J_{ijkl}$ gives the standard SYK hamiltonian.
The symmetric part of $J_{ijkl}$ yields a constant, since fermions anticommute: $\{\psi_i, \psi_j\} = \delta_{ij}$. Since all fermion operators disappear, the residual sum over $ijkl$ yields $z$. This is a purely analytic argument. Interestingly, eq. (4.8) is indeed the shift in the ground state energy, after cancelling $\arctan$ and $\tan$ and factors of $T$. So one does not need to perform a numerical fit to obtain a precise relation between $j_0$ and $z$(modulo subtleties of taking a $\log $ of a complex $Z$, presumably this is why you have $\arctan(\
tan)$ in eq. (4.8) ). I strongly believe that Section 6 should emphasize this point.
  • validity: -
  • significance: -
  • originality: -
  • clarity: -
  • formatting: -
  • grammar: -

Author:  Victor Godet  on 2022-02-02  [id 2143]

(in reply to Report 1 on 2022-01-22)

Thanks for your comment. I agree that $z$ can be interpreted as a coupling-dependent ground state energy. I have modified the end of paragraph below 6.2 to reflect this, reproduced here:

This quantity has the interpretation of a coupling-dependent ground state energy. For example, a simple way to change the value of $z$ is to shift the couplings by a fixed complex constant. Overall, this just adds a constant to the Hamiltonian after using the anticommutation relations of the fermions. Our purpose in isolating this simple parameter is to make a comparison with the gravity side. We will see that $z$ appears to be closely related with the zero mode $j_0$ of $j(\tau)$.

The pattern in the imaginary part of F is indeed a consequence of the branch cut in the argument. This effect results from the fact that the partition is a sum of two exponentials, so we have schematically

$$ \mathrm{Im}(F) = -T \,\mathrm{arg} \, Z,\qquad Z = e^{S_0} +e^{S_1} e^{ i j_0/T} $$
For $S_1> S_0$, i.e. when the half-wormhole dominates over the black hole, $Z$ circles around the origin as $T$ is changed and this is what gives the saw pattern. The dependence of $j_0$ is indeed that of a ground state energy. In gravity, this is the (complex) ground state energy of the half-wormhole. I believe that this distinctive behaviour is a hint that the SYK model at fixed couplings also contains the half-wormhole saddle-point, since such behaviour is most easily accounted by added the second oscillatory exponential in $Z$ (the contribution of the half-wormhole). Also let me point out, as we discuss in section 6, that the identification of $j_0$ and $z$ is mostly heuristic and based on our numerical results. To obtain a precise relationship, one would need to go beyond numerics in SYK.

Anonymous on 2022-02-06  [id 2161]

(in reply to Victor Godet on 2022-02-02 [id 2143])

I thank Authors for the quick response. I recommend the paper for publication.

---

## Round 2 · List of Changes

minor changes detailed in the replies to the referees

---

## Round 3 · Author Response

Minor revisions

---

## Editorial Decision

published